



# Optimising CH$_4$ simulations from the LPJ-GUESS model v4.1 using an adaptive MCMC algorithm

Jalisha T. Kallingal[1], Johan Lindström[2], Paul A. Miller[1], Janne Rinne[3], Maarit Raivonen[4], and Marko Scholze*[1]

[1]Department of Physical Geography and Ecosystem Science, Lund University, Lund, Sweden
[2]Centre for Mathematical Sciences, Lund University, Lund, Sweden
[3]Natural Resources Institute Finland
[4]Institute for Atmospheric and Earth System Research/Physics, Faculty of Science, University of Helsinki, Helsinki, Finland

**Correspondence:** Jalisha T. Kallingal (jalisha.theanutti@nateko.lu.se)

**Abstract.** The processes responsible for methane (CH$_4$) emissions from boreal wetlands are complex, and hence their model representation is complicated by a large number of parameters and parameter uncertainties. The arctic-enabled dynamic global vegetation model LPJ-GUESS is one such model that allows quantification and understanding of the natural wetland CH$_4$ fluxes at various scales ranging from local to regional and global, but with several uncertainties. The model contains detailed descriptions of CH$_4$ production, oxidation, and transport controlled by several process parameters.

Complexities in the underlying environmental processes, warming-driven alternative paths of meteorological phenomena, and changes in hydrological and vegetation conditions are highlight the need for a calibrated and optimised version of LPJ-GUESS. In this study we formulated the parameter calibration as a Bayesian problem, using knowledge of reasonable parameters values as priors. We then used an adaptive Metropolis Hastings (MH) based Markov Chain Monte Carlo (MCMC) algorithm to improve predictions of CH$_4$ emission by LPJ-GUESS and to quantify uncertainties. Application of this method on uncertain parameters allows greater search of their posterior distribution, leading to a more complete characterisation of the posterior distribution with reduced risk of sample impoverishment that can occur when using other optimisation methods. For assimilation, the analysis used flux measurement data gathered during the period 2005 to 2014 from the Siikaneva wetlands in southern Finland with an estimation of measurement uncertainties. The data are used to constrain the processes behind the CH$_4$ dynamics, and the posterior covariance structures are used to explain how the parameters and the processes are related. To further support the conclusions, the CH$_4$ flux and the other component fluxes associated with the flux are examined.

The results demonstrate the robustness of MCMC methods to quantitatively assess the interrelationship between objective function choices, parameter identifiability, and data support. As a part of this work, knowledge about how the CH$_4$ data can constrain the parameters and processes is derived. Though the optimisation is performed based on a single site's flux data from Siikaneva, the algorithm is useful for larger-scale multi-site studies for more robust calibration of LPJ-GUESS and similar models, and the results can highlight where model improvements are needed.



# 1 Introduction

$CH_4$ is the second most important long-lived greenhouse gas after carbon dioxide ($CO_2$) (Ciais et al. (2013); Kirschke et al. (2013)). It has been reported that the global atmospheric $CH_4$ concentration has been growing since the pre-industrial time.
In 2021 it reached a value of 1908 parts per billion (ppb), nearly 2.62 times greater than its estimated value in 1750 (Dlugo-kencky, 2021). This increase in the atmospheric concentration of $CH_4$ is responsible for around 16.5 % of the total effective radiative forcing (in $W\ m^{-2}$) of the well-mixed greenhouse gases (IPCC AR6: Forster et al. (2021)). Despite its relatively shorter turnover time of less than ten years in the troposphere, $CH_4$ has a much stronger infrared radiation absorption capacity compared to $CO_2$ (Prather et al., 2012).

Among the biogenic sources, wetlands contribute around 19-33% of current global terrestrial $CH_4$ emissions and are the largest and the most uncertain (Kirschke et al. (2013); Saunois et al. (2020); Bousquet et al. (2006)). Wetlands occupy around 3.8% of the Earth's land surface and are mainly located in high latitude regions. There is approximately 455 Pg of carbon stored in boreal and subarctic wetland peat/histosols which is accumulated by absorbing atmospheric $CO_2$ by plants as part of their photosynthesis (Gorham, 1991). Under long-term anaerobic soil situations, this carbon will be metabolised by the anaerobic microorganisms called methanogens and will eventually be emitted back as $CH_4$ to the atmosphere (Aurela et al., 2009).

In the future, climate change may cause a positive feedback on emissions from wetlands $CH_4$ due to a warmer and wetter climate (Johansson et al. (2006); Bridgham et al. (2008)). According to Zhang et al. (2017) at the end of the twenty-first century, 38-56% of the $CH_4$ production from the wetlands would be climate change induced. It is also expected to have increased uncertainty in $CH_4$ emission from boreal wetlands (Christensen et al., 2007) partly due to expected spatio-temporal changes in wetland extent (Saunois et al., 2016). Considering the fragility of boreal wetlands and the possibility that they fail to maintain their stability in a changing environment (Jacob et al., 2007), one way to quantify their carbon budget is to model their to model their carbon dynamics, including their $CH_4$ emission. Realistic and optimised process-based vegetation models can be used to reach a more precise estimation of emission variability and trends. However, representation of the complex biogeochemical processes, including soil carbon turnover, vegetation dynamics, hydrology, soil thermal dynamics, and defining wetland boundaries are complex, so, estimating the contribution from multiple pathways for $CH_4$ production, consumption, and release complicates wetlands $CH_4$ modelling (Melton et al. (2013); Wania et al. (2010, 2013); Susiluoto et al. (2018)), thus, different models represent these biogeochemical and biophysical processes differently with varying degrees of complexity.

The Lund-Potsdam-Jena General Ecosystem Simulator (LPJ-GUESS) (Smith et al., 2014) is one of a few available process-based dynamic global vegetation model (DGVM) that simulates local to global vegetation dynamics and soil biogeochemistry (Smith (2001); Sitch et al. (2003)). Taking the information about the climate and concentration of $CO_2$ in the atmosphere, it predicts the structural, compositional, and functional properties of the native ecosystems of major climate zones of the Earth. Considering the complexity of LPJ-GUESS with its large number of uncertain process parameters the model requires a mathematically robust framework for parameter optimisation (Wramneby et al., 2008). Data assimilation using Bayesian statistics can be seen as a way of combining observations with prior information (i.e. model process formulation and prior model parameter values) to derive posterior parameter and emission estimates (Susiluoto et al. (2018); Ghil and Malanotte-





Rizzoli (1991); Dee (2005); Carrassi et al. (2018)). The Markov Chain Monte Carlo (MCMC) (Metropolis et al., 1953b) is a powerful and convenient Bayesian framework (Tarantola, 1987) for data assimilation as it can combine prior information with observations to sample from the posterior distributions in complex models. This study has developed an Adaptive MCMC Metropolis Hasting (AMCMC-MH) framework (Hastings (1970b); Tarantola (1987)) with Rao-Blackwellised adaptation of

the multivariate Gaussian random walk proposals (Andrieu and Thoms, 2008). The algorithm minimises the model-data misfit, i.e. a cost function, by sampling from the probability density function (PDF) of the posterior parameters. The adaptation allows the algorithm to learn the shape of the posterior, improving sampling efficiency. The main objective of this paper is to evaluate the capabilities and limitations of the AMCMC-MH framework to optimise $CH_4$ wetland emissions simulated by the LPJ-GUESS model by analysing the posterior parameter distributions, the parameter correlations and the processes they control.

Considering the complexity of LPJ-GUESS with its large number of uncertain process parameters (Wramneby et al. (2008); Wania et al. (2010)), there is a need for a mathematically robust framework for parameter optimisation.

## 2 Data and Methodology

### 2.1 Siikaneva wetland and measurements

The Siikaneva wetland is located at 61° 49°N, 24° 11°E, at 160 $m$ a.s.l and is the second-largest un-drained wetland complex

in Southern Finland (Ahti et al. (1968); Rinne et al. (2007)). This boreal wetland complex has an area of 12 $km^2$, including minerotrophic and ombrotrophic sites with over 6 meters of peat deposition under the surface (Mathijssen et al. (2016); Aurela et al. (2007); Rinne et al. (2007)). The estimated average annual total precipitation is about 707 $mm$. The average temperature for January and July are approximately -7.2°$C$. and 17.1°$C$, respectively. The estimated mean annual temperature is around 4.2°$C$ (Korrensalo et al., 2018). The total annual $CH_4$ emissions from the Siikaneva wetland varies between 6.0 $gCm^{-2}$ and

14 $gCm^{-2}$ and net $CO_2$ fluxes vary between -96 $gCm^{-2}$ and 27 $gCm^{-2}$ (Rinne et al., 2018).

Daily measurement of incoming short wave radiation, precipitation, and air temperature collected at the wetland are used as input to the model. Since the meteorological data measured directly at the Siikaneva wetland have several significant gaps, which made them unsuitable as inputs to the model, we used precipitation and temperature data collected from a nearby station called Juupajoki-Hyytiälä (around 5.5 kilometres away from Siikaneva, open data by Finnish Meteorological Institute (FMI):

https://en.ilmatieteenlaitos.fi/download-observations) and the short wave radiation data collected from the Hyytiälä weather station (SMEAR II station around 6 kilometres away from Siikaneva, https://smear.avaa.csc.fi/download (Hari et al., 2013) ). Given the short distances between these sites and Siikaneva, we assumed that the meteorological variables are representative of Siikaneva. To verify the assumption, we have analysed the available data from Siikaneva and the datasets collected from Juupajoki and Hyytiälä sites. The air temperature and precipitation of the Juupajoki and the Siikaneva showed a Pearson cor-

relation of 0.998 and 0.706, respectively . Short wave radiation data collected at Hyytiälä and Siikaneva showed a correlation of 0.98. Still, there were some minor gaps in the short wave data collected at Hyytiälä, which were therefor gap-filled using the available data collected at Siikaneva for the corresponding periods. Additional inputs to the model are atmospheric $CO_2$ con-





centration as described by McGuire et al. (2001) and updated until recent years using data from the NOAA Global Monitoring Laboratory (https://gml.noaa.gov/ccgg/trends).

## 2.2 CH₄ model description in LPJ-GUESS

LPJ-GUESS contains a number of well-defined modules to represent the related ecosystem processes with a distinct spatial and/or temporal signature. Compared to version 4 of the model described by Smith et al. (2014), version 4.1 which we used for this study, has more detailed representations of plant functional types (PFTs) characteristics and processes in wetlands (Gustafson (2022)). This include improved descriptions of peatland-specific PFTs, peatland hydrology, soil temperature estimation, and CH₄ emissions. These process descriptions and developments (with some minor modification) were adopted from the wetlands and CH₄ module in the LPJ-WHyMe model (Wania et al. (2009a, b, 2010)), and are described in detail in McGuire et al. (2012). Brief descriptions of the important wetland processes in LPJ-GUESS version 4.1 are given below, for more detailed description see Gustafson (2022).

### 2.2.1 Active peat column and properties

The active wetlands peat in the LPJ-GUESS is represented by a 1.5 *m* deep column further divided into 15 layers of 0.1 *m* thickness each (see Figure 1 ). The uppermost three layers comprise the acrotelm, within which the water table can vary. The underlying 12 layers of catotelm are are saturated with water permanently Wania et al. (2009a); Gustafson (2022).

The acrotelm layers have a porosity ($por_{acro}$) of 0.98, while the catotelm layers, assumed to be made of older, denser peat, have a porosity ($por_{cato}$) of 0.92. Each layer consists of constant proportions of peat and varying proportions of water ($F_{water}$), ice ($F_{ice}$), and air ($F_{air}$), all with distinct thermal characteristics given in Table 1. The active column is covered by a maximum of five snow layers, with a depth that can reach 10 m water equivalent, and five extra padding layers that extend to a depth of 48 m. These layers are thermally active, but, hydrologically inactive, with the bottom three layers having thermal properties of bedrock (Table 1).

### 2.2.2 Peat temperature

Temperature in each active peat layer is calculated daily by solving the heat diffusion equation;

$$\frac{\partial T}{\partial t} = \frac{\partial}{\partial x}\left(D(z,t)\frac{\partial T}{\partial z}\right) \tag{1}$$

where $T$ represents the temperature of the soil at a a specific depth z (*m*) and time *t*, while *D(z,t)* $(m^2 s^{-1})$ denotes the thermal diffusivity at depth z and time *t*, defined as:

$$D(z,t) = \frac{K(z,t)}{C(z,t)} \tag{2}$$



**Table 1.** Heat capacities ($10^6\,Jm^{-3}K^{-1}$) and thermal conductivities ($Wm^{-1}K^{-1}$) of the soil layer components. The values are originally adopted from, Wania et al. (2009a, b); Bonan et al. (2002); Granberg et al. (1999); and Chadburn et al. (2015).

| Component | Heat capacity | Thermal conductivity |
|-----------|---------------|----------------------|
| Peat | 0.58 | 0.06 |
| Water | 4.18 | 0.57 |
| Ice | 1.94 | 2.2 |
| Air | 0.0012 | 0.025 |
| Bedrock | 2.1 | 8.6 |

where $K(z,t)$ ($W\,m^{-1}K^{-1}$) represents the thermal conductivity, and $C(z,t)$ ($Jm^{-3}K^{-1}$) represents the soil layer component's heat capacity (Table 1), each at a depth z and time *t*, more details can be seen in Wania et al. (2009a, b).

Water plays a major role in the wetland's soil temperature because of the dynamics of latent heat during its phase change (Wania et al. (2009a, b)). When temperature changes over time and depth, T(z,t) in the soil, the values of $F_{water}$ and $F_{air}$ also change due to phase change, with a similar spatial (0.1 *m*) and temporal (1 *day*) resolution.

The calculation for freezing and thawing of water in version 4.1 of LPJ-GUESS is different from that described in Wania et al. (2010). It calculates them below the wilting point and freezing of the water stored above the wilting point can occur only after all the water below the wilting point has frozen. Likewise, melting of the ice stored above the wilting point can only take place once all the ice below the wilting point has melted.

### 2.2.3  Peat hydrology

The hydrology of acrotelm layers follow the description of Wania et al. (2009a, b) originally following Granberg et al. (1999). As mentioned above, it is assumed that the catotelm layers remain saturated permanently with no inflow or outflow, but to maintain saturation, water is added to these layers on a daily basis, if necessary. This is because PFTs such as graminoid species can absorb water from the catotelm layers via their roots.

Thus, the model updates only the daily water content in acrotelm, and predicts the water table depth ((wtd)), where $0 <=$
(wtd) $<= 300$ *mm*, i.e. (wtd) is positive below the surface, and standing water is not permitted. Each day the change in total volume of water in acrotelm (V) is calculated as:

$$\Delta V = run_{on/off} + rain_{melt} - evap - aet_{acro} - runoff_{acr0} \tag{3}$$

where *evap* represents the amount of water that gets evaporate from the bare peat soil fraction, $rain_{melt}$ represents the daily amount of water input to the patch as rainfall and/or snowmelt, $runoff_{acro}$ is the runoff from the acrotelm and $aet_{acro}$ is the





transpiration from the acrotelm based on the root distributions in the acrotelm layers. The user has the option to include a site-specific *runon/off*, which enables them to mimic local conditions by either adding (*runon/off* $> 0\ m$) or removing (*runon/off* $< 0\ m$) water from the acrotelm, if they are known.

Once the total volume of water is determined, the water table depth (*(wtd)*) in the acrotelm is assumed to be linear in the first top interval (0-0.1 $m$) and constant below this depth and up to the lower limit of the acrotelm, i.e. 0.1-0.3 $m$ (Granberg et al., 1999). Hence if 0.1 $m >=$ *(wtd)* $>= 0$ the *(wtd)* is calculated as:

$$wtd = \sqrt{\frac{3\,(por_{acro} \times 0.3 - V)}{2 \times a_z}} \tag{4}$$


And for *(wtd)* $> 0.1\ m$ the *(wtd)* is calculated as:

$$wtd = \frac{1.5 \times (por_{acro} \times 0.3 - V)}{por_{acro} - f_{surfmin}} \tag{5}$$

where $f_{surfmin} = 0.00025$ is the surface minimum fractional water content in $m^3/m^3$, $por_{acro}$ is the porosity in the acrotelm and $a_z = por_{acro} - f_{surfmin}/0.1$ is the gradient in the uppermost 0.1 $m$ suction interval, The water profile of soil $\theta(z)$ in each 150 layer of 0.1 $m$ is calculated as,

$$\theta(z) = min(por_{acro}, \theta_{surf} + (por_{acro} - \theta_{surf}) \times \left(\frac{z}{wtd}\right)^2) \tag{6}$$

where the $\theta_{surf}$, the surface water content is calculated as,

$$\theta_{surf} = max(f_{surfmin}, por_{acro} - wtd \times a_z) \tag{7}$$

     Once $\theta(z)$ in each 0.01 $m$ layer is known, the average of ten 0.01 $m$ layers is used to calculate the fractional water content 155 ($F_{water}$) in each of the three 0.1 $m$ sublayers of the acrotelm, which will then used for calculating the thermal properties, i.e. for the soil temperature calculations described above.

### 2.2.4   Peatland PFTs

Table 2 provides the properties of four types of PFTs that can exist on peatland stands. The descriptions of *Sphagnum mosses* and *C3 graminoids* in model is taken from Wania et al. (2009b). The model includes a generic herbaceous cushion lichen moss 160 PFT (pCLM), low deciduous and evergreen shrubs (*pLSE* and *pLSS*, respectively). Both of these PFTs are parameterized to favor dry peatlands that have low water tables.







**Figure 1.** Schematic representation of the $CH_4$ model in LPJ-GUESS coupled with the CENTURY soil organic model. Carbon for methanogens is allocated to soil layers based on the distribution of roots in each layer. The root density decreases from top to bottom of peat. The assigned carbon in each layer is divided into $CH_4$ and $CO_2$. Oxygen ($O_2$) either directly diffuses or is transported through plants. The availability of $O_2$ determines the amount of $CH_4$ in the soil as it oxidises a fraction of $CH_4$. Similarly $CH_4$ also can either directly diffuse or be transported to the atmosphere in bubbles, or it can be transported by vascular plants. The equilibrium between gaseous bubbles of $CH_4$ and dissolved $CH_4$ in water is controlled by the maximum solubility of $CH_4$. Any $CH_4$ that exists in gaseous form will escape to the atmosphere via ebullition.

The leaf area index (LAI) of nearby trees or shrubs is a limiting factor on PFTs. The model sets a maximum LAI limit of 2 $m^2 m^{-2}$ for mosses and graminoids, and exceeding this limit leads to increased shade mortality.

Similarly a daily desiccation stress factor [0,1] and an inundation stress factor are also introduced in the model. A desiccation
stress factor of 1 indicates that there is no stress, whereas a value of 0 signifies complete suspension of photosynthetic activity for that day (applies only to mosses and graminoids). Inundation stress factor is implemented to control assimilation when the





**Table 2.** Important parameter values used for defining Wetland PFTs. Here the $WTD_{inun}$ (*mm*) is the maximum *(*wtd) threshold, and $inund_{days}$ (*days*) that are the number of days wetland PFTs can tolerate inundated conditions.

| PFT | $WTD_{inun}$ | $inund_{days}$ | Aerenchyma | Photosynthesis stress due to lower *(*wtd) |
|---|---|---|---|---|
| pLSE, pLSS | 250 | 5 | No | N/A |
| Sphagnum moss | 50 | 15 | No | 0.3 |
| C3 graminoids | N/A | N/A | Yes | 0.0 |
| pCLM | 200 | 10 | No | N/A |

rooting zone experiences anoxia. The model restricts PFTs with a maximum *(*wtd) threshold and the number of inundated days ($inund_{days}$) they can tolerate before assimilation (Table 2).

### 2.2.5 SOM dynamics and daily decay rates

The Soil organic matter (SOM) scheme in the LPJ-GUESS is adopted from the CENTURY model (Parton (1996); Smith et al. (2014)) with eleven distinct pools of different carbon : nitrogen (C : N) stoichiometry and base decay rates (Figure 1). The decomposition rates in the acrotelm, which is often wet and sometimes saturated, are slow. In the catotelm, where conditions are permanently saturated and anaerobic, decomposition rates are particularly slow (Frolking et al. (2001, 2010)). The decomposition rate for wetlands is computed daily for each pool using the following equation:

$$\frac{dC_j}{dt} = -k_{j,max}f(T)f(W)f(S).C_j \tag{8}$$

where $C_j$ is the carbon content in pool $j$, $k_{j,max}$ is maximum decay rate, $f(T_{soil})$, $f(W)$ (from here on-wards called $Rmoist$) and $f(S)$ are dimensionless scalars between $0 - 1$ related to soil temperature, soil moisture and soil fractional silt plus clay content ($S$) respectively. Considering the negligible soil fractional silt plus clay content in peat ($S = 0$), $f(S) = 1$.

From the parameter sensitivity test conducted by Wania et al. (2010), the value of $Rmoist$ in LPJ-GUESS is adopted as 0.4 180 for carbon in the acrotelm. After the acrotelm soil carbon is fully established, which involves a peat layer 0.3 m deep with a carbon density of 25 *kg C* $m^{-3}$, corresponding to a total soil carbon amount of 7.5 *kg C* $m^{-2}$ across all pools, the value of $Rmoist$ will be reduced from the weighted average 0.4 to 0.025, hence in anaerobic catotelm conditions the moisture response $Rmoist_{anaerobic} = 0.025$; following Ise et al. (2008) and Frolking et al. (2001, 2010).

### 2.3 CH₄ dynamics in high-latitude wetland stands (above 40° latitude)

The decomposed organic carbon in each day (explained in Section 2.2.5) is distributed vertically in different peat soil layers weighted by an assumed static root distribution, exponentially declining from the surface to the deeper layers, see Equation 9. In high-latitude wetlands, this carbon pool is considered as 'potential carbon pool' for methanogenic archaea, and is the basic concept behind the CH₄ model in LPJ-GUESS. The total available carbon is decomposed into two components, $CO_2$ and $CH_4$



depending on the availability of $O_2$ in the soil. The dissolved $CH_4$ concentration and the gaseous $CH_4$ fraction are calculated
based on the estimated $CH_4$ content in each layer. A portion of the estimated $CH_4$ is oxidised by the soil $O_2$ and the remaining
is transported to the atmosphere by either diffusion, ebullition, or plant-mediated transport. Apart from being the key factor
in estimating the 'potential carbon pool', root biomass in each soil layer also plays a role in the transport of $O_2$ and $CH_4$ into
and out of each layer is mediated by plants. From different studies of various wetland PFTs Wania et al. (2010) observed an
exponential decrease of root biomass with depth proportional to the degree of anoxia, which is expressed by the following
equation, also used in LPJ-GUESS;

$$f_{root} = C_{root} e^{z/\lambda_{root}} \tag{9}$$

where $f_{root}$ is the fraction of root biomass at a certain depth z, $\lambda_{root}$ = 0.2517 $m$ is the decay length and $C_{root} = 0.025$ is a
normalisation constant. This distribution ensures that approximately 60% of the roots are distributed within the acrotelm, and
the root fraction in the lowest soil layer is adjusted to achieve a total root distribution of 1 across all 15 soil layers.

### 2.3.1 CH$_4$ production

Due to its wide ranges, the $CH_4$/$CO_2$ ratio from decomposition is a challenging task to predict. For example, Segers (1998)
observed a high variation in the molar ratio of $CH_4$ to $CO_2$ production between 0.001 to 1.7 in anaerobic conditions. Hence it
is taken in the model as an adjustable parameter weighted by the degree of anoxia $\alpha$, determined as $\alpha$ = 1-($F_{air}$+$f_{air}$), where
$F_{air}$ is the fraction of air in the soil layers and $f_{air}$ is the fraction of air in peat (Wania et al., 2009a) (see the Section 2.2.1 for
details).

The production of $CH_4$ in each day in each layer is determined as,

$$\text{CH}_{4prod} = \alpha(z) \times f_{root}(z) \times CH_4/CO_2 \times R_h \tag{10}$$

where $\alpha(z)$ is the degree of anoxia at depth z, $f_{root}$(z) is the fraction of root in the peat at depth z, $CH_4/CO_2$ =0.085 (in the
model), is the tuning parameter for the $CH_4$ to $CO_2$ production ratio and $R_h$ is the daily heterotrophic respiration. Note that the
model is set to $\text{CH}_{4prod}$ = 0 when $F_{water}$<0.1, assuring zero $CH_4$ production in frozen and/or dry soils, i.e, the model assume
there is no water when the water is frozen, hence $F_{water}$ is 0.

### 2.3.2 CH$_4$ oxidation

The $CH_4$ fraction that is oxidised depends on the availability of $O_2$ (represented by the parameter $f_{oxid}$= 0.5 in the model) in
the soil. A part of the $O_2$ transported to the soil will be consumed by the plant roots and non-methanotrophic microorganisms.
The remaining part is then used to oxidise $CH_4$. The oxidised $CH_4$ is added to the $CO_2$ pool, and the remainder stays in the
$CH_4$ pool and will get transported at each time step.





### 2.3.3 Total CH$_4$ flux

Diffusion, ebullition and plant-mediated transport are the three pathways through which CH$_4$ is transported to the atmosphere.
The total CH$_4$ flux from high-latitude wetland patches in the model is represented as,

$$F_{\text{CH}_4} = \text{CH}_{4diff} + \text{CH}_{4plant} + \text{CH}_{4ebul} \tag{11}$$

where CH$_{4diff}$ is the CH$_4$ flux component from diffusion, CH$_{4plant}$ is the CH$_4$ flux component from plant-mediated transport and CH$_{4ebul}$ is the CH$_4$ flux component from ebullition. Since the daily CH$_4$ production in each layer is dependent on $R_h$ (Equation 10), $F_{\text{CH}_4}$ is subtracted from $R_h$ before saving it. Any CO$_2$ generated, whether from heterotrophic respiration or
CH$_4$ oxidation, is released into the atmosphere.

**Diffusion**

The fractions of CH$_4$, CO$_2$ and O$_2$ that are transported to the atmosphere and from the atmosphere through diffusion are calculated by solving the gas diffusion equation within the peat layers using a Crank-Nicolson numerical scheme with a time step of 15 *minutes*. The molecular diffusivities of these gases in soil depend on temperature, soil porosity and the water and air
contents in the soil. Diffusivity in water is derived by fitting a quadratic curve to observed diffusivities at different temperatures as described in Broecker and Peng (1974); diffusivity in the air and its temperature dependency is derived from the values taken from Lerman et al. (1979), and diffusivity in soil and its temperature dependency is estimated from the Millington and Quirk model described in Millington and Quirk (1961). A detailed description can be seen in Wania et al. (2010).

At the water-air surface the gas diffusivities changes by minimum four orders of magnitude, hence at the water-air boundary,
the flux is calculated by the following equation,

$$J = -\psi(C_{surf} - C_{eq}) \tag{12}$$

where $C_{surf}$ is the surface water gas concentration, and $C_{eq}$ is the concentration of gas in equilibrium with the atmospheric partial pressure, estimated using Henry's law. $\psi$, the gas exchange coefficient, also called piston velocity, is usually difficult to estimate for different gases. In this case, the piston velocities of CH$_4$, CO$_2$ and O$_2$ are calculated by relating them to the known
piston velocity of SF$_6$ by the following equation,

$$\psi* = \psi_{600}\left(\frac{Sc*}{600}\right)^n \tag{13}$$

where $\psi_{600} = 2.07 + 0.215 \times U_{10}^{1.7}$ is the piston velocity of SF$_6$ normalised to a Schmidt number of 600 (subjected to the wind speed $U_{10}$ at 10 *m* from the ground, which is considered as zero in the model), Sc* represents the Schmidt number of the gas under consideration, and n = - 1/2. See Wania et al. (2010) for details.





As mentioned above the diffusion through the soil is affected by soil porosity, hence by the value of $F_{air}(t,z)$. When $F_{air}$ $\leq 0.05$ in soil layers the diffusivities in water are used. When $F_{air} > 0.05$, the diffusivities in air, which are four orders of magnitude larger than those in water, become more significant. For soil layers where $F_{air} \leq 0.05$, the diffusivities in water are used. When $F_{air} > 0.05$, the diffusivities in air, which are four orders of magnitude larger than those in water, become more important. Each day before diffusion is calculated, the gas flux J at the boundary is used to update the dissolved gas

content. The surface concentration $C_{surf}$ of CH$_4$ will mostly be greater than $C_{eq}$; hence J will be negative, denoting flux to the atmosphere, though it is possible for CH$_4$ to diffuse into the soil in small amounts if the concentrations at the surface are suitable. The resulting daily flux of CH$_4$ is determined as the total CH$_{4diff}$.

**Ebullition**

"Ebullition depends on the solubility of CH$_4$ at a given temperature and pressure and occurs when the water table reaches the
surface during periods of high CH$_4$ emission. Following Wania et al. (2010), in LPJ-GUESS, the best-fitted curve is represented as;

$$S_B = 0.05708 - 0.001545T + 0.00002069T^2 \tag{14}$$

where $S_B$ is the Bunsen solubility coefficient, i.e. the volume of gas dissolved per volume of liquid at atmospheric pressure and a given temperature (Wania et al., 2010).
The CH$_4$ in each layer is converted to a maximum allowable dissolved mass, and this limit is used to separate the CH$_4$ in the form of dissolved and gaseous components. If there is any CH$_4$ that exceeds the maximum solubility of a layer, it will be released into the atmosphere. The CH$_{4ebul}$ is calculated by adding this ebullition fluxes from all layers.

**Plant-mediated transport**

Plant-mediated transport of CH$_4$ occurs via the aerenchyma (the gas-filled tissues) of vascular plants either through concentra-
tion gradient or active pumping from soil to the atmosphere. Only the passive mechanism (through concentration gradient) is considered in the model as it is the most dominant one (Cronk and Fennessy, 2016). Abundance, biomass, phenology and the rooting depth of aerenchymatous plants are considered to calculate this. Only the flood-tolerant $C_3$ graminoid is considered for plant-mediated gas transport in the model (Table 2); hence plant-mediated transport of O$_2$ and CH$_4$ can only occur when $C_3$ graminoids are present in a simulated patch.
The transport depends on the cross-sectional area of plant tillers[1] in each soil layer, assuming that a significantly high percentage of CH$_4$ is oxidized in the highly oxic zone near the roots, where methanotrophs flourish, before they enter into the plants tissue.

The mass of their tiller is calculated as,

---

[1]Tiller refers to all the secondary shoots produced by grasses (Poaceae or Gramineae). Each tiller stem is segmented with its own two-part leaf.





$$m_{tiller} = b_{graminoid} \times P(leaf) \tag{15}$$

where $b_{graminoid}$ is the leaf biomass of graminoids, and 'P' represents the daily phenology, which is the fraction of potential leaf cover that has reached full development. To calculate number of tillers ($n_{tiller}$) total weight of tillers, $m_{tiller}$, is devdied by the average weight of an individual tiller ($w_{tiller}$). The cross-sectional area of tillers, $A_{tiller}$ then can be obtained by,

$$A_{tiller} = n_{tiller} \times \phi_{tiller} \times \pi r_{tiller}^2 \tag{16}$$

where $r_{tiller}$ is the tiller radius and $\phi_{tiller}$ is the tiller porosity. Based on the optimisation of McGuire et al. (2012), Tang et al.
(2015) and Zhang et al. (2013) the value of $r_{tiller}$ is estimated as 0.0035 *m* and based on the Wania et al. (2010), the values of $\phi_{tiller}$ and $W_{tiller}$ are estimated as as 70% and 0.22 *gC/tiller* respectively. Each soil layer is allocated a fraction of the total cross-sectional area of tillers based on the root fraction estimated in that layer. The $CH_{4plant}$ is estimated by adding the plant-mediated $CH_4$ fluxes from all layers.

### 2.4 Parameters selected for optimisation

Parameter values related to the processes of $CH_4$ emission in LPJ-GUESS are mostly adopted from the parameter values described in Wania et al. (2010). Since Wania et al. (2010) had difficulties finding the optimal parameter values for many of the parameters, they performed some preliminary analysis for seven uncertain parameters, for which there were little or no data available. They performed a simple initial sensitivity test by taking four sets of values for each of the seven parameters, followed by a parameter fitting exercise with three sets of values for every seven parameters. They ran the model with all their
2187 different combinations for seven sites for one year. As a result, they got a Root-mean-square error (RMSE) range between 226.4 and 18.3 (*mg* $CH_4$ $m^{-2}$ $d^{-1}$ ) for the different sites, which clearly indicates loosely fitted parameters with a high degree of uncertainty.

    In this study, parameters for the optimisation are selected based on their sensitivity to the model output ($CH_4$) and expert opinion. We used a simple method to calculate the percentage difference in output (single simulation) when varying only one
input parameter at a time from its permitted minimum value to its maximum (Hoffman and Miller (1983); Bauer and Hamby (1991)). The 'sensitivity index' (SI) is calculated using the equation,

$$SI = \frac{D_{max} - D_{min}}{D_{max}} \tag{17}$$

where $D_{min}$ and $D_{max}$ represent the model output values corresponding to the minima and maxima of the corresponding parameter range.


We considered five of the seven parameters Wania et al. (2010) tested in their sensitivity analysis (two parameters related



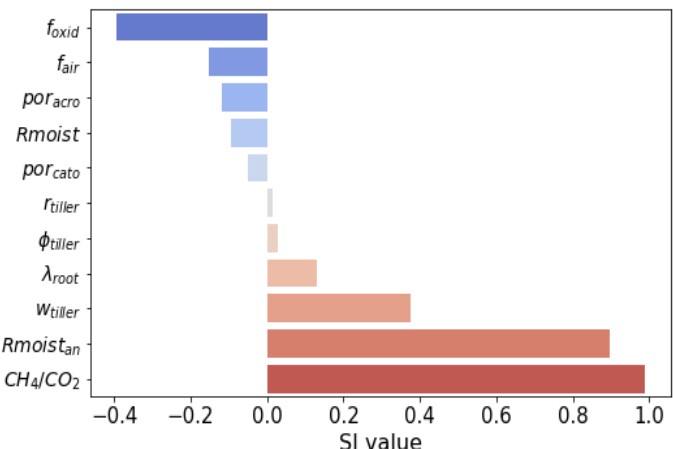

**Figure 2.** Selected parameters for the optimisation and their SI values. The red and blue colours indicate the increase and decrease in total CH$_4$ flux, respectively, when the value of the parameter increases.

to the root exudate decomposition are not used in LPJ-GUESS) together with six other parameters used in LPJ-GUESS based on their high SI values (Figure 2, Table 3).

Among the selected parameters $Rmoist$ and $Rmoist_{anaerobic}$, the response of soil organic matter decomposition to the

soil moisture content in acrotelm and catotelm conditions respectively (Equation 8); CH$_4$/CO$_2$ the CH$_4$ to CO$_2$ ratio in the anaerobic conditions (Equation 10); $f_{air}$, the fraction of air in peat (Section 2.3.1 and Equation 10); $por_{acro}$ and $por_{cato}$ the porosity in acrotelm and catotelm respectively (Section 2.2.1); $\lambda_{root}$, the decay length of root biomass in peat (Equation 9) are the parameters related to the CH$_4$ production. The $f_{oxid}$, fraction of available O$_2$ used for CH$_4$ oxidation, (Section 2.3.2) is the parameter related to the CH$_4$ oxidation. $w_{tiller}$, the average weight of an individual tiller; $r_{tiller}$, the tiller radius and $\phi_{tiller}$ the

tiller porosity (Equation 16) are the parameters related to the CH$_4$ transportation.

### 2.5  Parameter optimisation framework

After selecting the parameters to be optimized and the physical possible range of values for each parameter, we assumed Gaussian probability density functions (PDF) to depict both the prior distributions of the parameters and the deviation between model and observations. The resulting model can be formulated as,

$$Y|x \sim N(M(x), R),$$
$$x \sim N(x_p, B), \tag{18}$$

where $Y$ are the observations, $M(x)$ is the LPJ-GUESS output given parameters $x$, $x_p$ is the prior values of the parameters, and $R$ and $B$ are error covariance matrices describing the uncertainty in observations and priors, respectively.

The prior uncertainties, $B$, are based on expert opinion and were kept relatively large to reduce the prior's influence on the

posterior parameter estimates. We have assumed prior variance for each parameters as 40% of their expected range, see Table





**Table 3.** Selected parameters for the assimilation related to the CH$_4$ flux from LPJ-GUESS. Prior values, prior standard deviation (std), units, and description used for the prior distribution are given.

| Number | Parameter | Prior value | Prior std | Unit | Description |
|---|---|---|---|---|---|
| 1. | $Rmoist$ | 0.4 | 0.396 | - | Moisture response in acrotelm |
| 2. | CH$_4$/CO$_2$ | 0.085 | 0.236 | - | Anaerobic CH$_4$ to CO$_2$ ratio |
| 3. | $f_{oxid}$ | 0.5 | 0.36 | - | Litter CO$_2$ fraction |
| 4. | $\phi_{tiller}$ | 70 | 36 | % | Tiller porosity in percentage |
| 5. | $r_{tiller}$ | 0.0035 | 0.004 | $m$ | Tiller radius in meter |
| 6. | $f_{air}$ | 0 | 4 | % | Fraction of air in peat |
| 7. | $por_{acro}$ | 0.98 | 0.06 | - | Porosity in catotelm |
| 8. | $por_{cato}$ | 0.92 | 0.076 | - | Porosity in enters slow soil carbon pool |
| 9. | $Rmoist_{an*}$ | 0.025 | 0.04 | - | Moisture response in catotelm |
| 10. | $w_{tiller}$ | 0.22 | 0.24 | $gC$ | Tiller weight in gram carbon |
| 11. | $\lambda_{root}$ | 25.17 | 12 | $cm$ | Decay length of root biomass in centimeter |

3. The parameters are also assumed to be a prior uncorrelated, due to lack of good and consistent expert opinions regarding dependence.

### 2.5.1 Cost Function

Using the Bayesian framework the posterior for the parameters becomes

$$P(x|Y) = \frac{P(Y|x)p(x)}{p(Y)} \propto P(Y|x)p(x),\tag{19}$$

which in log-scale results in the quadratic loss function as (Tarantola, 1987)

$$\log P(x|Y) = -J(x) + \text{const.}$$
$$J(x) = \frac{1}{2}(Y - M(x))^t R^{-1}(Y - M(x)) + \frac{1}{2}(x - x_p)^t B^{-1}(x - x_p)\tag{20}$$

where const. represents normalising constants not depending on the unknown parameters. The two terms in $J(x)$ represent data-model misfit and the prior information on the parameters. A number of experiments aim to achieve the smallest cost function values to locate the optimal parameter set within the parameter space.

### 2.5.2 Adaptive Metropolis-Hastings

To search for the optimal posterior parameters, we used a MCMC-MH algorithm (Metropolis et al. (1953a); Hastings (1970a)). The algorithm generates samples from a target distribution by, in each iteration, drawing from a proposal distribution and then either accepting the new state or copying the old state. The resulting sequence of states will represent dependent samples from the target distribution.





Tuning the proposal distribution is important for obtaining an efficient sampling from the target distribution. A badly tuned MH algorithm will result in poor or incomplete convergence of the sequence and slow mixing, i.e. the sequence will take very long to produce samples from the correct distribution. Manual tuning of the proposal distribution is often time-consuming and prone to errors, especially for complex non-linear models, such as LPJ-GUESS, which can be sensitive to initial values

and have complex posterior distributions with multiple local minima. Instead we used an adaptive scheme where the MCMC MH automatically learns features of the target distribution (Andrieu and Thoms (2008), Roberts and Rosenthal (2009)). We call the resulting framework the *Global Rao-Black-wellised Adaptive Metropolis* (G-RB AM) algorithm, since it combines the Rao-Black-wellised Adaptive Metropolis algorithm with the Global Adaptive Scaling Metropolis algorithm, both described in Andrieu and Thoms (2008). See Supplement S1 for technical details.

## 345 2.6 Experiment design

### Twin experiment

A simple twin experiment is designed to assess the performance of the developed G-RB AM and its ability to recover the parameter values. The daily $CH_4$ output simulated by the LPJ-GUESS using randomly chosen true parameter values ($Z_{true}$) within their permitted range of variation are used as the synthetic observation. Since the synthesized observation conforms

completely to the model, any potential errors in the model or uncertainties in observations have not influenced the parameter optimization process, ensuring unbiased posteriors. It is expected that the assimilated parameters converge to the $Z_{true}$ values when the MCMC chain progress in time. To freely recover the $Z_{true}$ values, the prior parameter value ($x_p$) in the cost function (Equation 20) is set as $Z_{true}$. Two scenarios are considered for the twin experiment to test the identifiability of the parameters under different conditions. Scenario 1 with a shorter temporal scale from 2005 to 2014 (10 years); scenario 2 with a longer

temporal scale from 1901 to 2015 (115 years). Scenario 1 is more realistic and is chosen to mimic the real data at Siikaneva, whereas scenario 2 constitutes an ideal, hypothetical case with observations over the entire simulation period. Four sets of chains for both scenarios with a chain length of 100,000 iterations are analysed. In each set of the scenarios, the optimisation started from a different initial point in parameter space randomly selected from their prescribed ranges.

### Real Data experiment

To estimate the posterior parameter values, an experiment with a chain length of 100,000 iterations using the real observation from Siikaneva is designed. The observed daily averages are compared with the model simulation in the cost function only when more than 90% of the hourly observation were available each day. When there are gaps in the daily observation, we eliminate them, and their corresponding modelled values from the cost function calculation. In principle the error covariance matrix R should include both observation uncertainties and their correlations. From the fact that the latter is difficult to estimate,

we neglected them, and the observation uncertainties are estimated as 30% for the daily observations greater than 0.01 $gC$ $m^{-2}d^{-1}$, and a floor value of 0.3 for the observations less than 0.01 $gC\,m^{-2}d^{-1}$.





## 2.7 Parameter value estimation

For all the experiments conducted in this study, the first 75% of the G-RB AM chains are discarded as the 'burn-in'. The PDFs generated after the 'burn-in' are used to estimate each parameter's maximum a posteriori probability (MAP), posterior
mean and standard deviation (std). Following the idea used in Braswell et al. (2005) the parameter distributions are grouped into three categories: 'well-constrained', 'poorly constrained', and 'edge-hitting' parameters. The well-constrained parameters are the ones that exhibit a well-defined uni-modal distribution, with low std. The poorly constrained parameters are the ones that exhibit a relatively flat multi-modal distribution with large std. To be more precise with the estimation, for the posterior parameter distributions appeared multi-model if the std of the distribution is greater than 20% of its total range, we classified
them as poorly constrained. The edge-hitting parameters are the ones that cluster near one of the edges of their prior range (Braswell et al., 2005).

## 2.8 Posterior re-sampling experiment

To examine the effect of parameter optimisation on flux components, we designed a re-sampling experiment from the posterior parameter distributions. From the experiment conducted using site observation, 1000 sets of parameters are randomly selected
and used to run the model to simulate the $CH_4$ flux components. The outputs from each simulation of the experiment are used to analyse the process correlations and process-parameter relationships.

## 3 Results

## 3.1 Twin experiment using G-RB AM

The trace-plot resulted from the four different twin experiments of scenario 1 is illustrated in Figure S2:1 (see supplemental
information). The result of scenario 2 is not shown, as it also followed the same pattern. The Figure shows the convergence of each chain to the $Z_{true}$ values regardless of their chosen initial values. The result shows a good convergence of all parameters except the $CH_4/CO_2$ and $\lambda_{root}$. Posterior parameter correlations of the experiment 1 shown in Figure S2:1 are given in S2:2. Among the poorly retrieved parameters six of them except $\phi_{tiller}$ are observed as weak positively correlated to each others. The $\phi_{tiller}$ showed a weak negative correlation to $Rmoist$ but positive correlation to all the other poorly retrieved parameters. The
resulting PDFs of the experiment 1 after the 'burn-in' are represented in Figure 3. This figure shows the mean and MAP values as well as the std of the parameters; their numerical values are given in Table 4. In general twin experiments have resulted in 'well-constrained' and 'poorly constrained' parameter classes. Examples of the different classes of the distributions for the experiment 1 of scenario 1 are shown in Figure 3. Based on the posterior distributions estimated from all the four G-RB AM chains the parameters $Rmoist$, $CH_4/CO_2$, $f_{oxid}$, $r_{tiller}$, $f_{air}$, $por_{acro}$, $por_{cato}$ and $\lambda_{root}$ are well constrained in scenario
1 and the parameters $Rmoist$, $CH_4/CO_2$, $f_{oxid}$, $r_{tiller}$, $f_{air}$, $por_{acro}$, $por_{cato}$, $w_{tiller}$ and $\lambda_{root}$ are well constrained in scenario 2 (Table 4).



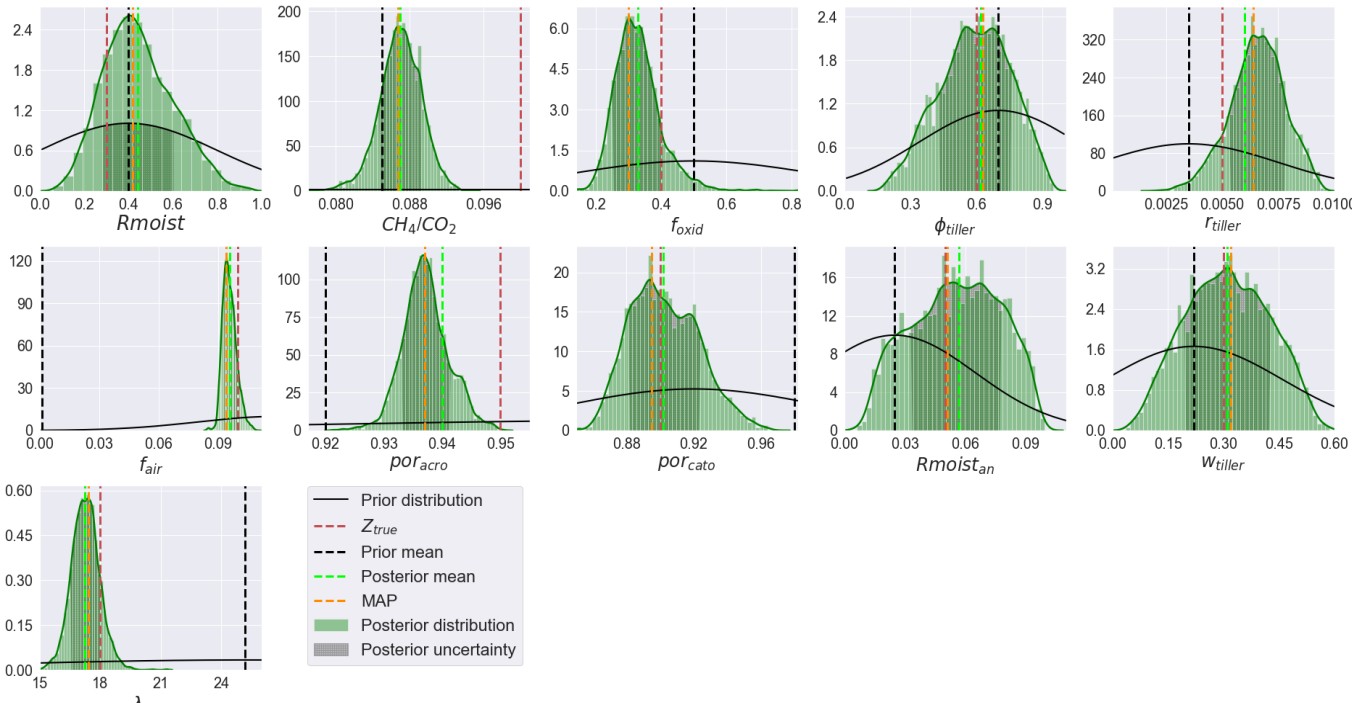

**Figure 3.** An example of probability density functions (PDFs) from the twin experiment, after the 'burn-in'. Prior and posterior distributions are illustrated with black and green solid lines respectively. True parameter values ($Z_{true}$), prior mean, posterior mean and maximum a posteriori probability (MAP) are shown in red, black, lime and orange colors respectively. Parameter behavior: $Rmoist$, $CH_4/CO_2$, $f_{oxid}$, $r_{tiller}$, $f_{air}$, $por_{acro}$, $w_{tiller}$ and $\lambda_{root}$ are well constrained (WC); $\phi_{tiller}$, $por_{cato}$ and $Rmoist_{ana}$ are poorly-constrained (PC).

The parameter retrieval capacity of the G-RB AM algorithm is estimated as the 'retrieval score' by dividing the posterior mean estimates of the parameters from all the chains in each scenario by $Z_{true}$ parameter values. The idea behind the retrieval score is that in an ideal case of complete recovery, the posterior parameter estimate and the $Z_{true}$ value are the same; hence the retrieval score would be one. Figure 4 shows the retrieval scores obtained for each parameter and their $1\sigma$ value. In scenario 1 the $\phi_{tiller}$, $por_{acro}$, $por_{cato}$, $w_{tiller}$ and $\lambda_{root}$ are well retrieved with a low std. Scenario 2 performed better in parameter retrieval compared to scenario 1, in which the majority of the parameters except the $CH_4/CO_2$, $r_{tiller}$, $w_{tiller}$ and $\lambda_{root}$ were showing good retrieval scores, but with comparatively high stds (see Figure 4). The overall mean retrieval score estimation is based on the ratio of the estimated and true values, given a value of 0.95 with a std of 0.19 for scenario 1, and a value of 1 with std 0.21 for scenario 2 (see Figure 4), which is again an indication for the good performance of the G-RB AM.

The reduced posterior cost function values and their $\chi^2$ values are given in Table 5. Here the reduced $\chi^2$ values are calculated by dividing twice the cost function by the number of observations used in the assimilation. Overall the $\chi^2$ values indicate a statistical robust cost function reduction given the prescribed uncertainties. The comparatively smaller values of $\chi^2$ for sets 1 and 3 in scenario 1 and set 3 in scenario 2 indicates a tendency to over fitting the results and being overconfident in the



**Table 4.** Means, standard deviations (std) and maximum a posteriori probability (MAP) of retrieved parameters for selected twin experiments in both scenarios and the parameter classes estimated from analysing the distributions of all four chains. The parameter classes include well-converged (WC) and poorly-converged (PC) parameters.

| | | **Parameter** | | | | | | | | | | |
| | | $Rmoist$ | $CH_4/CO_2$ | $f_{oxid}$ | $\phi_{tiller}$ | $r_{tiller}$ | $f_{air}$ | $por_{acro}$ | $por_{cato}$ | $Rmoist_{ana}$ | $w_{tiller}$ | $\lambda_{root}$ |
| Sc 1 | $Z_{true}$ | 0.30 | 0.1 | 0.40 | 0.60 | 0.005 | 0.10 | 0.95 | 0.90 | 0.05 | 0.30 | 18.0 |
| | MAP | 0.42 | 0.086 | 0.31 | 0.63 | 0.006 | 0.094 | 0.93 | 0.89 | 0.051 | 0.32 | 17.4 |
| | Posterior mean | 0.40 | 0.087 | 0.33 | 0.60 | 0.006 | 0.096 | 0.94 | 0.90 | 0.057 | 0.30 | 17.2 |
| | $std \mp$ | 0.16 | 0.002 | 0.07 | 0.16 | 0.001 | 0.003 | 0.004 | 0.02 | 0.02 | 0.10 | 0.70 |
| | Class | WC | WC | WC | PC | WC | WC | WC | WC | PC | PC | WC |
| Sc 2 | $Z_{true}$ | 0.30 | 0.1 | 0.40 | 0.60 | 0.005 | 0.10 | 0.95 | 0.90 | 0.05 | 0.30 | 18.0 |
| | MAP | 0.22 | 0.079 | 0.24 | 0.64 | 0.006 | 0.09 | 0.94 | 0.89 | 0.064 | 0.29 | 13.9 |
| | Posterior mean | 0.28 | 0.08 | 0.26 | 0.63 | 0.006 | 0.10 | 0.95 | 0.89 | 0.053 | 0.26 | 14.1 |
| | $std \mp$ | 0.07 | 0.002 | 0.05 | 0.18 | 0.001 | 0.0008 | 0.0009 | 0.006 | 0.01 | 0.01 | 0.42 |
| | Class | WC | WC | WC | PC | WC | WC | WC | WC | PC | WC | WC |

**Table 5.** Cost function reduction observed from the G-RB AM twin experiments using two different scenarios (Sc). Prior and posterior cost function values obtained from four sets of experiments for each scenario are given. The misfit of observed and expected (zero) cost function values are represented as the reduced $\chi^2$ value.

| | Experiment | Prior | Posterior | $\chi^2$ |
|---|---|---|---|---|
| Sc 1 | Set 1 | 12486.4 | 301.6 | 0.17 |
| | Set 2 | 49674.0 | 759.6 | 0.422 |
| | Set 3 | 29535.6 | 294.0 | 0.17 |
| | Set 4 | 8476.8 | 428.0 | 0.24 |
| Sc 2 | Set 1 | 86140.0 | 6170.0 | 0.31 |
| | Set 2 | 619172.0 | 8040.0 | 0.38 |
| | Set 3 | 68792.0 | 3372.4 | 0.16 |
| | Set 4 | 109888.0 | 8646.0 | 0.41 |

estimated posterior values and uncertainties.

## 3.2 Real data experiments and optimised parameters

For the experiment with the real data, the observations collected at the Siikaneva wetland are assimilated using the G-RB AM algorithm. The trace plots with 100,000 iterations obtained after the optimisation are exemplified in Figure 5.

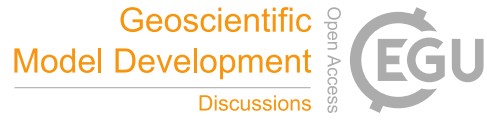

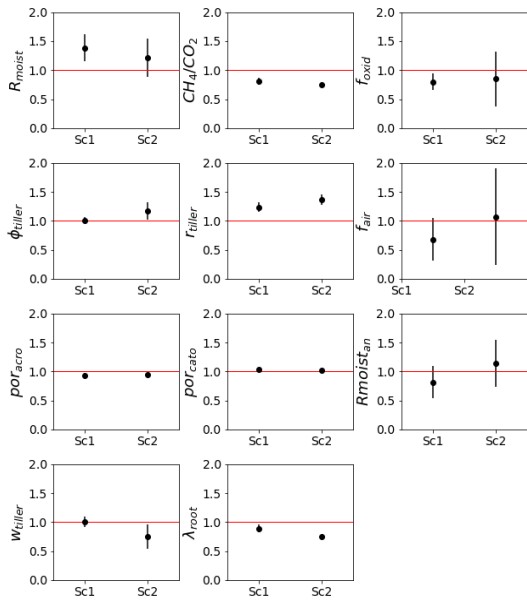

**Figure 4.** Twin experiments result in terms of mean retrieval score based on the ratio of the estimated and $Z_{true}$ values of the parameters. The horizontal red lines indicate a complete retrieval, and the error bar shows the std from different chains in different scenarios.

### 3.2.1 Optimised parameter values and distributions

The posterior parameter PDFs are shown in Figure 6. The shapes of the distributions are used to interpret the results of the parameter optimisation as explained in Section 2.5. In contrast to the twin experiments, the parameters fell into three categories: 'well-constrained', 'poorly constrained', and 'edge-hitting'; the classifications are given in Table 6. The PDFs for parameters $Rmoist$, $CH_4/CO_2$, $\phi_{tiller}$, $f_{air}$, $por_{acro}$, $w_{tiller}$ and $\lambda_{root}$ are classified as well constrained distributions. The PDFs for $r_{tiller}$, $por_{cato}$ and $Rmoist_{anaerobic}$ are classified as poorly constrained distributions, and the one for $f_{oxid}$ is classified as a edge-hitting distribution. Both in the well-constrained and poorly constrained parameters, high kurtosis is observed. The values of $f_{oxid}$, which is the edge-hitting parameter, lay near the higher bound of the edges of the prior range, and most of the retrieved values were clustered near this edge. The parameter also exhibited large positive kurtosis and negative skewness. Apart from their shapes, the MAP and the posterior mean estimates were also computed. The estimated posterior parameter values and their $1\sigma$ stds along with the prior values are shown in Table 6. The MAP and posterior mean estimates of the parameter agree on the value for $CH_4/CO_2$, $f_{air}$ and $por_{acro}$. For $f_{oxid}$, $\phi_{tiller}$ and $r_{tiller}$, both the MAP and posterior mean estimates stayed out of 1/3 of the $1\sigma$ range of the posterior distribution, which we consider a large difference, and for the remaining parameters, the MAP and posterior mean estimates stayed within 1/3 of the $1\sigma$ of their posterior distribution; hence we consider this as a small difference.

For the parameters $Rmoist$ and $CH_4/CO_2$ the posterior values appeared very close to, but slightly below the prior values. The posterior values of $Rmoist_{anaerobic}$ appeared very close to, but slightly above the prior values. For the parameter $\phi_{tiller}$

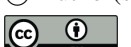

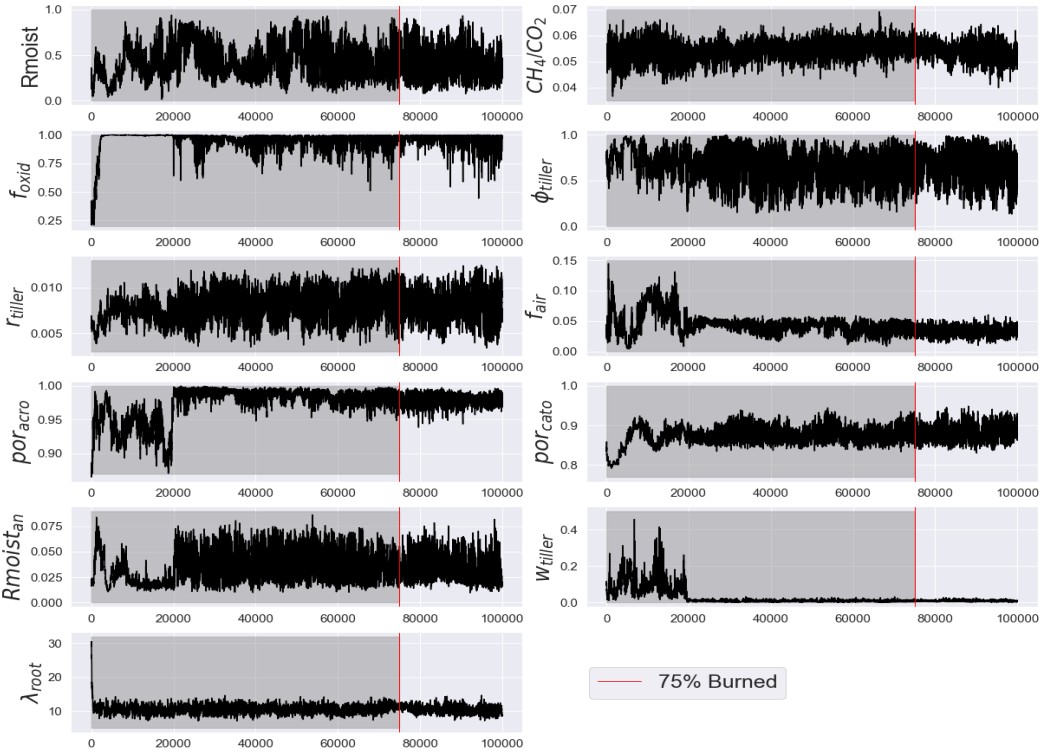

**Figure 5.** An example of the G-RB AM chains for the experiment with real observations showing all the 10,000 values in the chain. The first 75 % was discarded as 'burn-in' and is greyed out in the figures. The remaining 25% (from the red vertical lines) is used for the analyses.

the MAP estimate appeared very close to but above the prior value and posterior mean estimate appeared very close to but below the prior value. For these four parameters, the posterior mean stayed within 1/3 of the $1\sigma$ range of the assumed prior uncertainty. The parameters $f_{oxid}$, $r_{tiller}$, and $f_{air}$ posterior values appeared slightly above the prior values, but out of the 1/3

of the $1\sigma$ range of the prior uncertainty. The prior and posterior values of the parameter $por_{acro}$ remained the same. In contrast, parameters $por_{cato}$, $w_{tiller}$ and $\lambda_{root}$ appeared far distant from, and below the prior values, out of 1/3 of the $1\sigma$ range of prior uncertainty, but stayed within the prior range (see Section 4.2 for details).

### 3.2.2 Posterior parameter correlation

The 2D distributions of the posterior parameters and their Pearson correlations are illustrated in Figure 8. Overall, the majority

of the parameters showed weak positive or negative correlations with a few exceptions with extreme correlations (the values and corresponding colour code in the triangle above depict this). For example $Rmist_{anaerobic}$ showed high negative correlation to $R_{moist}$ and $por_{acro}$ showed high positive correlation to the $f_{air}$. The 2D marginal distributions (scatter plots), illustrated in the lower triangle, showed a general tendency of high clustering within the $1\sigma$ range for all the parameters; in general, the 1D





histograms (on the diagonal, also shown in Figure 6) appeared as well-constrained uni-modal distributions. For further details,
see Section 4.3.1.

### 3.2.3   Cost function reduction

The prior parameter values and cost function value, as well as the posterior parameter values and cost function values corresponding to both posterior MAP and mean estimates are listed in Table 6. The prior cost function value calculated with the default model parameters showed a high-cost value of 48424.4 with a model overestimation of around four times the observed
flux. After the optimisation, the cost function value was reduced to 2959.8 with the MAP estimate of parameters and to 3002.6 with the posterior mean estimate of parameters.

As anticipated, the cost function was marginally lower for the MAP estimate when compared to the posterior mean estimate, resulting in a better model-data fit regarding the error model with the MAP estimate, which can be seen in Figure 9b. It can also be observed from the Figure that the cost function reduction has not only fitted the total model sum to the total observational
sum but also has reduced the misfit between each year.

### 3.2.4   Flux components of CH$_4$ simulation and parameter values

To understand how and how much in magnitude each optimised parameter influences the flux components and the total flux, the result of the 're-sampling experiment' (see Section 2.8) is examined by correlation and regression analyses. The Pearson correlation coefficients and regression slopes are calculated for all the 1000 parameter sets and their corresponding total sums
of the flux components and total flux. The left side of Figure 7 shows a schematic summary of the correlation coefficients and regression slopes between the 11 parameters and the flux components including total flux. For the total flux, all parameters except fro $f_{oxid}$ and $\phi_{tiller}$ showed a similar regression pattern observed in the case of diffusion with slight differences in magnitudes. This similarity is not surprising as diffusion is the most dominant process among the process components. The total flux showed highest correlations to $CH_4/CO_2$ and $\lambda_{root}$ and lowest correlation to $f_{oxid}$. A detailed discussion of the
process-parameter relations can be found in Section 4.3.2.

The right side of Figure 7 shows the correlations between the sums of flux components resulted from the 're-sampling experiment'. The 2D distributions in the lower triangle show a strong positive relation between diffusion and total flux. Almost all the parameter residuals are observed within the $3\sigma$ deviation without many outliers. Except for the correlation between diffusion and total flux, the analysis showed no other strong positive or negative correlation between the components, as can
be seen in the correlation plot illustrated in the top triangle.





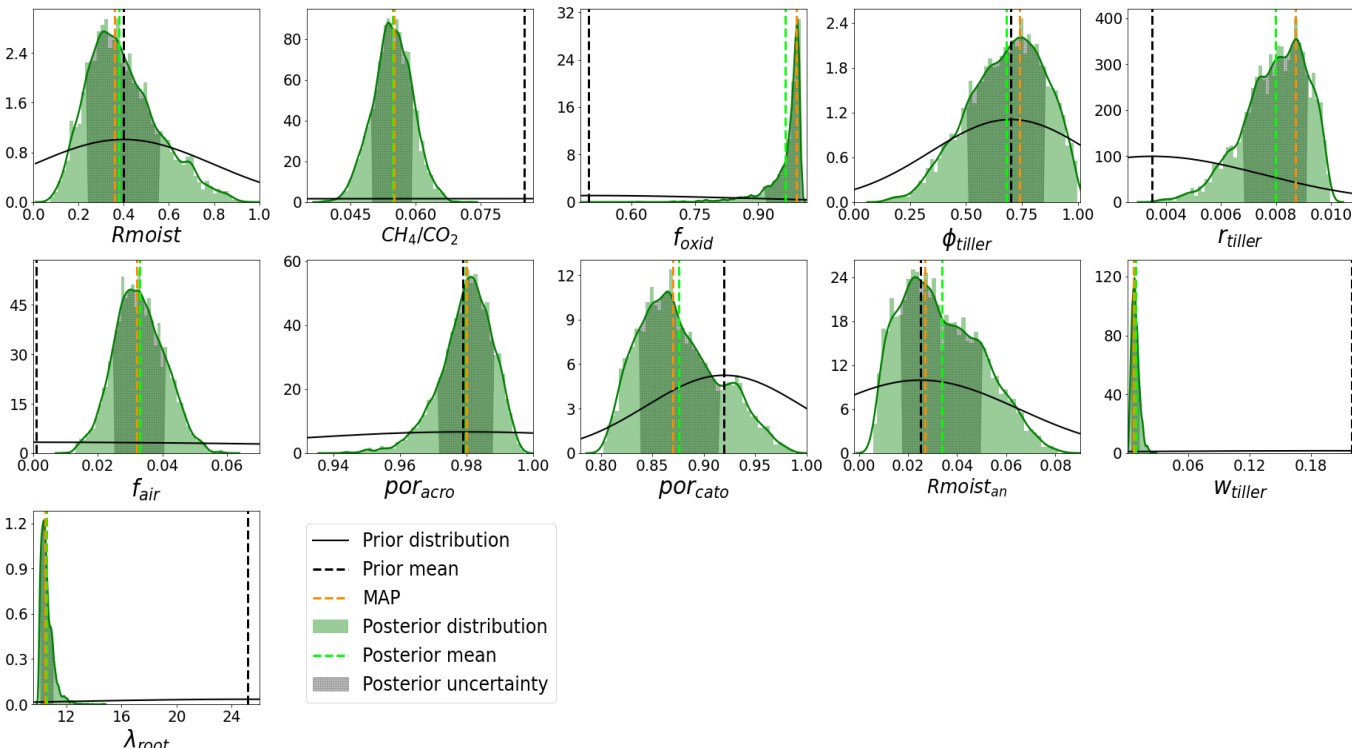

**Figure 6.** PDFs of parameters from G-RB AM real data experiment after the 'burn-in'. The green curves shown are the smoothed Gaussian kernel estimates of the posterior distribution on the posterior histograms, and the black curves are the prior distributions. The dotted vertical green and black lines are the posterior and prior means, respectively. The shaded green area of the distributions represents the $1\,\sigma$ error estimate of the PDFs.

Yearly variations in fractional contributions of flux components simulated using prior and posterior parameter estimates are examined to understand the impact of the optimisation on the composition of the inter-annual emissions. The time series of the annual sums of flux components as a function of their total flux (in percentage) are shown in Figure 9a. The result shows that among the flux components, diffusion contributes the most to the total $CH_4$ flux both in prior and posterior estimates, with a
slightly higher contribution in the posterior estimate, followed by plant-mediated transport. But in the case of the prior estimate, the diffusion contributed comparatively less for the first two and the last years compared to the remaining years. In contrast to this, the contribution of plant-mediated transport was high for these first two and last years. The observed contribution of diffusion is very low in the case of the posterior (see Section 4.3.4 for a more detailed discussion).

The time series model-observation mismatch of prior and posterior estimates for the annual total fluxes can be seen in Figure
9b; the values are in percentage of the observed $CH_4$ flux. The prior estimate showed a mismatch of around 600% for the first two years. Also, a considerably high mismatch is observed in the years 2011, 2012 and 2014. The MAP estimate remained near zero, while the posterior mean estimate exhibited a slightly negative values indicating an underestimation of the flux.





**Table 6.** Parameter values obtained after the G-RB AM real data optimisation. The prior values, maximum a posteriori (MAP), posterior mean, std and parameter classes are shown. The parameter classes include well-converged (WC) and poorly-converged (PC), and edge-hitting (EH) parameters. The cost function values correspond to the parameter values obtained with prior, MAP and posterior mean estimates are also shown.

| | Parameter | | | | | | | | | | | |
| --- | --- | --- | --- | --- | --- | --- | --- | --- | --- | --- | --- | --- |
| | $Rmoist$ | $CH_4/CO_2$ | $f_{oxid}$ | $\phi_{tiller}$ | $r_{tiller}$ | $f_{air}$ | $por_{acro}$ | $por_{cato}$ | $Rmoist_{an}$ | $w_{tiller}$ | $\lambda_{root}$ | **Cost value** |
| Prior values | 0.4 | 0.085 | 0.5 | 0.7 | 0.0035 | 0.0 | 0.98 | 0.92 | 0.025 | 0.22 | 25.17 | 48424 |
| MAP | 0.37 | 0.055 | 0.98 | 0.74 | 0.0087 | 0.032 | 0.98 | 0.87 | 0.029 | 0.0061 | 10.47 | 2959.8 |
| Posterior mean | 0.39 | 0.055 | 0.96 | 0.68 | 0.0079 | 0.032 | 0.98 | 0.88 | 0.033 | 0.0082 | 10.58 | 3002.6 |
| std $\mp$ | 0.15 | 0.0046 | 0.046 | 0.17 | 0.0011 | 0.007 | 0.008 | 0.038 | 0.016 | 0.0037 | 0.45 | |
| Class | WC | WC | EH | WC | PC | WC | WC | PC | PC | WC | WC | |

Interestingly, the MAP followed the same pattern as the prior estimation by showing an increase whenever the prior increased and a decrease whenever the prior decreased; however, the posterior mean estimate did not show this relation.

The fraction of the annual errors of the flux components of the total flux (in %) is shown in Figure 10. The effect of optimisation on the individual contributions of each component can be seen from the annual means (solid dots) of their fractional contribution to the total flux. The error bars represent the $1\sigma$ stds from the mean values. Among the prior estimates of flux components, the prior plant-mediated transport showed the largest error (22.5%), and the ebullition showed the smallest error (9.1%). In the MAP estimate, ebullition showed the highest error with a value of 12.3%, followed by diffusion and ebullition

with around the same value of error, 6.9% and 6.8%, respectively. For the estimate using posterior mean values, diffusion and plant-mediated transport showed around the same errors, 7.5% and 7.4%, and the ebullition showed the least error (2.6%). On the right-hand side of the figure, the fourth column displays the mean and errors for the inter-annual variation of the total fluxes obtained by prior parameter values and posterior estimates. The prior total estimate showed an error of 4.2%, and the mean and MAP showed an error of 0.66% and 0.72%, respectively.

### 3.3    Fit to the observation

Figure 9b illustrates the percentage model-data misfit, and Figure 11 shows the time series of the assimilated observations together with the model prior and posterior estimates with their uncertainties. As expected, the posterior estimate fitted the observations better than the prior estimate. The total RMSE estimated between the prior and observations were 0.044 $gC$ $m^{-2}$ $d^{-1}$, which got reduced to a value of 0.023 $gC$ $m^{-2}$ $d^{-1}$ for the posterior case. The result indicates that most of the

mismatch between the prior model estimates and observations was contributed by the large overestimation in the initial years.



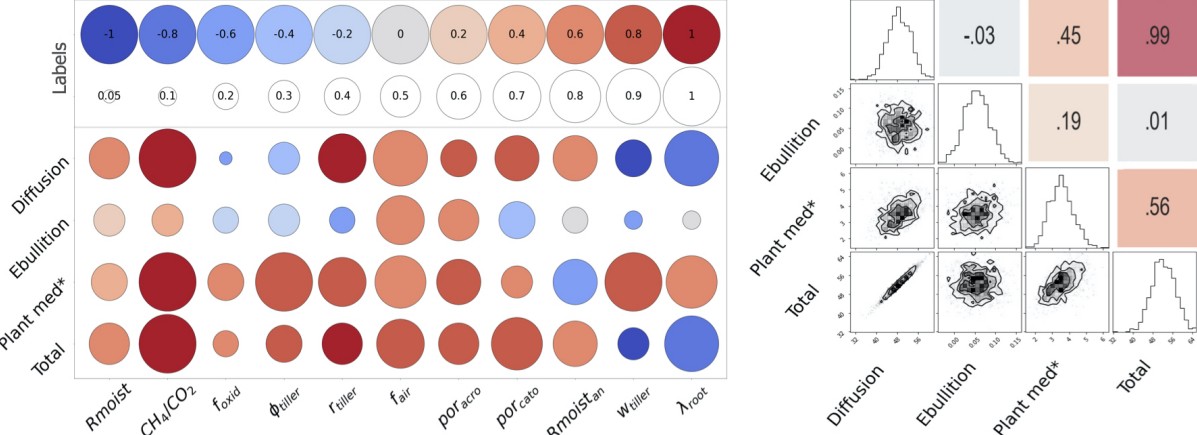

**Figure 7.** Schematic summery of the 're-sampling experiment'. The image on the left side shows the process-parameter correlation and regression slope. Three different flux components of $CH_4$ together with the total flux are labelled on the vertical axis, and the parameters are labelled on the horizontal axis. The different colours of the circles represent the regression slopes scaled between -1 and 1 (in 11 steps). The blue colour indicates a steeper negative slope hence a strong decrease, and the red colour indicates steeper positive slopes hence a strong increase in $CH_4$ fluxes with the increasing parameter value. The correlation coefficient ($R^2$) scaled between 0.05 and 1 (in 11 steps) is represented by the size of the circles, with larger circles indicating higher ($R^2$) values. The image on on the right side shows the process-process correlations. Numeric labels on the upper triangle correspond to Pearson's correlation coefficient values. The diagonal of the matrix shows the 1-D histogram for each flux components and the total flux. 2-D marginal distributions of the sum of the processes and total flux are represented in the lower triangle with contours to indicate $1\sigma$, $2\sigma$ and $3\sigma$ confidence levels. The points in the plots indicates the sums of flux components (black dots). Ranges of the distributions are labelled on the left and bottom of the figure.

This overestimation disappeared in the posterior, showing a better agreement with the observation. There are years for which the observations show large peaks during the summer (such as 2010, 2013 and 2014), and the posterior estimates succeeded in capturing these peaks to a large extent, see Section 4.6 for details.

# 4 Discussion

## 4.1 Twin experiment

A common problem with the adaptive MH algorithm is its inability to widely explore the target distribution if the set-up is not well tuned. This can then result in a poor approximation of the target distribution, hence poor adaptation. The resulting trace plots shown in Figure 5 and Figure S2:1 (see supplemental information) depict a set of well-explored parameters on their permitted space ranges during the progression of the random walk, which indicates a well-tuned assimilation framework. The use of the Blackwellised learning (as explained above) of the posterior distribution appeared beneficial during the transients



**Figure 8.** A posteriori correlations between the parameters from the G-RB AM real data optimisation. The blue and red colour in the upper triangle represents the strong negative and positive correlations, respectively. The numerical labels on the upper triangle are the values of Pearson's correlation coefficient. The panels on the diagonal show the 1-D histogram for each model parameter with a dashed red vertical line to indicate the best-fit value. The vertical blue lines are the 0.16, 0.5 and 0.84 quantiles of the distributions, respectively. On top of each 1D histogram, the mode of the distribution and the interval of the 0.16 and 0.84 quantiles are indicated. The lower triangle represents the two-dimensional marginal distributions of each parameter with contours to indicate $1\sigma$, $2\sigma$ and $3\sigma$ confidence levels, and the points in the plots are the values of G-RB AM chain after the 'burn-in' (blue dots). Ranges of the distributions are labelled on the left and bottom of the figure.





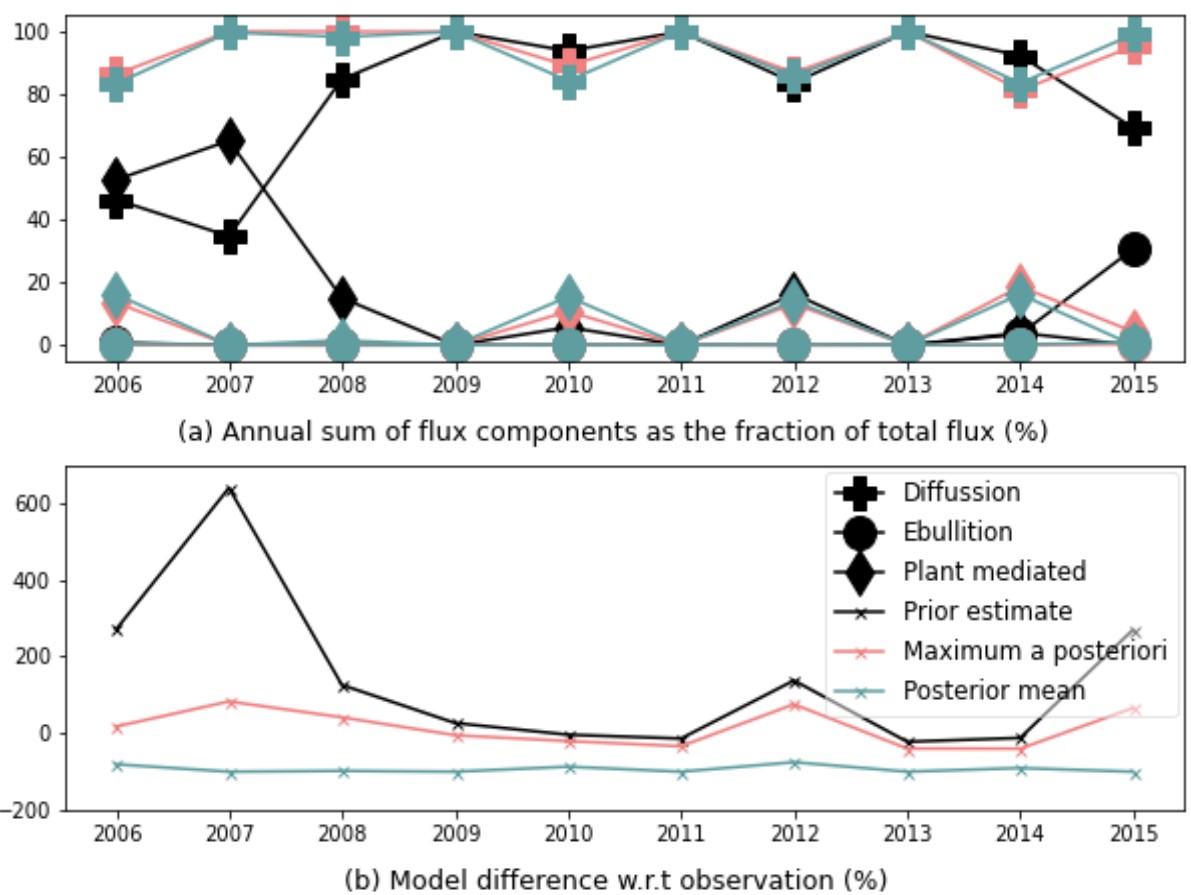

**Figure 9.** Flux component fractions and percentage model-data difference. Figure (a) shows the proportions of annual flux components plotted as a function of the total yearly flux. The different flux estimates are represented in solid lines of different colours, and the symbols on them correspond to each flux component. Figure (b) shows the annual model–observation mismatch in percentage with respect to the yearly total $CH_4$ observation

.

of the chains whenever the acceptance probability has been dropped to low values at low probability regions of the parameter space.

Figures 4 and Figure S2:1 show almost complete convergence of some parameters to $Z_{true}$ regardless of the scenarios. Given the complexity and non-linearity of the model, it is not surprising that not all parameters converged completely. It
is also not surprising that different chains estimated slightly different posterior solutions for the parameters. However, most poorly retrieved parameters still have their true values within the 1 $\sigma$ range of the Gaussian PDFs of the optimized values. The analysis of the cost function reduction (Table 5), the ability to constrain the parameters (Figure 3), and the parameter retrieval ability (Figure 4) of the twin experiments showed that the developed G-RB AM algorithm is capable of optimizing the process

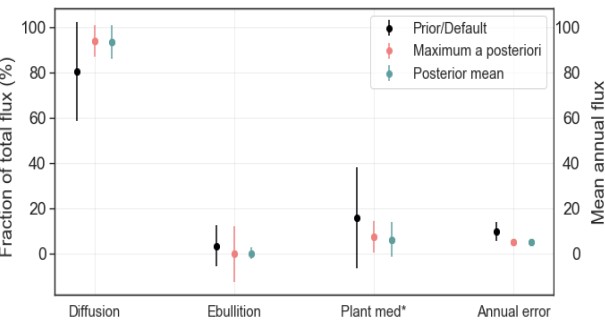

**Figure 10.** The first three columns of the figure show the fractions of the annual fluxes from process components of the total fluxes. The vertical solid lines represent the $1\sigma$ error bars of each component, and the dots represent the mean of the annual fluxes. The fourth column (correspond to y axis on the right side) shows the annual mean and annual errors for the inter-annual variation of the total fluxes.

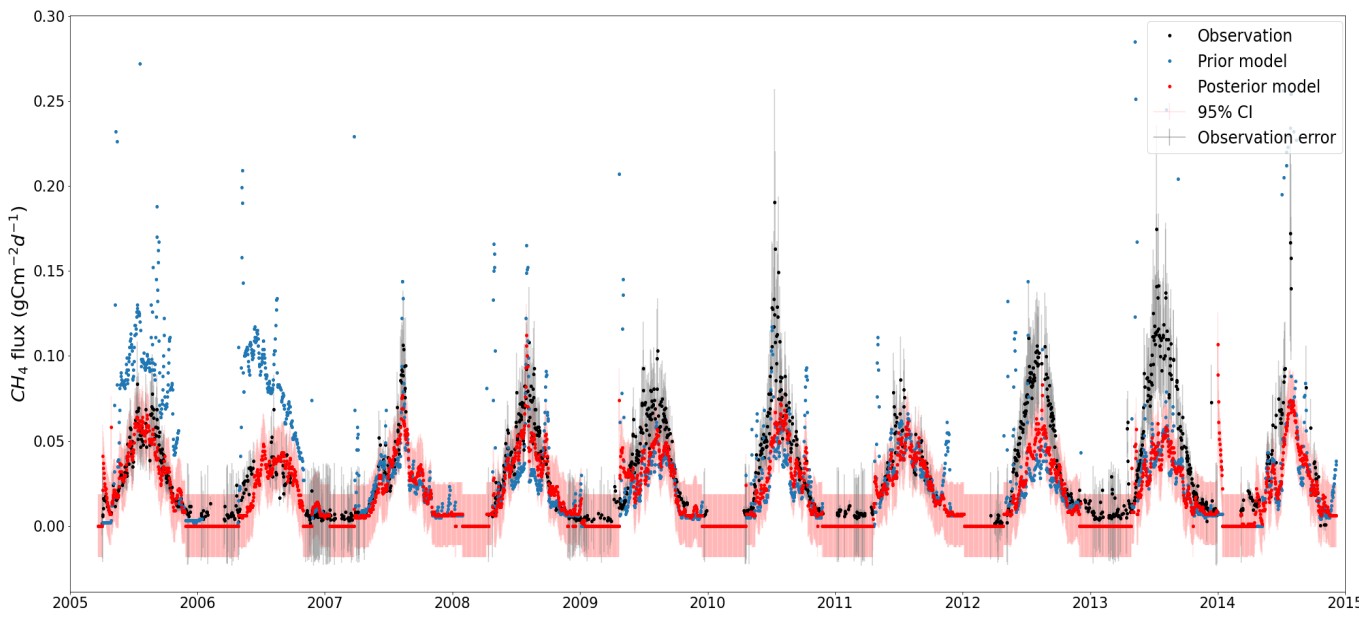

**Figure 11.** Total $CH_4$ simulation from the LPJ-GUESS model (red dots) after optimising with the G-RB AM algorithm. The black dots are the real $CH_4$ observations from Siikaneva with prior observation error (grey shade). The light red shade around the posterior model simulation is the 95% confidence interval ($CI$) of the simulations. The blue dots are the prior simulation with the prior default model parameters. A few outliers above 0.3 $gCm^{-2}$ on the vertical axis have been removed from the figure for better visualisation. While most of the observations fall within the confidence intervals, it's important to note that the effects of parameter variations in the posterior are part of these confidence intervals.





parameters related to $CH_4$ emissions in LPJ-GUESS. The results from the eight experiments conducted assuming observations

from two different scenarios indicate the capability of the algorithm for parameter retrieval regardless of the initial guesses and temporal scales used. The resulting posterior PDF distributions, characterized as uni-modal distributions, illustrate the ability of the developed framework to solve the multi-dimensional problem of reducing a complex cost function based on a highly non-linear model.

## 4.2    Parameter estimation using real observations

As described in Section 3.2.1 the experiment using real data resulted three poorly constrained and one edge-hitting parameter. The poorly constrained or edge-hitting parameters, however, are not uncommon in MH parameter search and rather expected with a complex and highly non-linear model such as LPJ-GUESS. The correlation of parameters to other parameters can affect the result; i.e. the number of parameters that can optimised within this data assimilation framework is limited. Though the twin experiments showed good parameter retrieval and non-equifinality, assimilating the complex real-world observations into

a complex ecosystem model like LPJ-GUESS is expected to have parameter retrieval and equifinality problems. This is one of the reasons for selecting a small subset of the parameters associated with wetland $CH_4$ flux simulations for this study. As described in Section 3.2.1 considerable changes have occurred to the prior parameter values after optimisation. Here it should be considered that, in general, while optimising the parameters, the assimilation is trying to reduce the $CH_4$ flux to minimise the misfit with the observed data, which is around half of the prior model estimate (see Table 7).

The very slight reduction, i.e. within 1/3 of the $1\sigma$ error in the posterior mean estimate of $Rmoist$ indicates a slight decrease of the moisture response in aerobic conditions, hence a slightly reduced $CH_4$ emission. Unlike $Rmoist$, the posterior mean estimate for $Rmoist_{anaerobic}$ got a higher value compared to the prior value, with a slight asymmetric multi-modal distribution. The higher posterior value of $Rmoist_{anaerobic}$ indicates the production of CH4 in the anaerobic conditions of the catotelm. The $CH_4/CO_2$ parameter, which is the $CH_4$ to $CO_2$ ratio in an anaerobic environment, was found to be lower as compared

to the prior. This indicates a high fraction of $CO_2$ production from the peat compared to $CH_4$ production. The prior parameter value for $f_{air}$ was zero, which means there is no 'permanent' gas fraction in peat. After the optimisation, the posterior value for $f_{air}$ was slightly positive (0.032), indicating a small air fraction in the peat. The high value of $f_{oxid}$ and $f_{air}$ which indicates a high available air fraction and/hence $O_2$ concentration in the soil to convert the available carbon into $CO_2$ respectively, could explain this reduction in $CH_4/CO_2$ as a balancing effect (Equation 10).

Among the $CH_4$ transport-related parameters, a slight reduction was observed in the posterior mean estimate of $\phi_{tiller}$, which indicates slightly more compact tillers with less porosity to transport the $CH_4$. A considerable reduction, more than 1/3 of the prior uncertainty, is observed in $w_{tiller}$, which indicates low leaf biomass. A decrease in the fraction of potential leaf cover would lead to a reduction in the amount of carbon added to the 'potential carbon pool' for methanogens, which will cause low $CH_4$ emission. Contradictory to the values of the two above-mentioned $CH_4$ transport-related parameters, $r_{tiller}$, which is the

tiller radius of plants, showed a value of more than twice the prior value, means more tillers for a given biomass, i.e. $n_{tiller}$ increases, which means $A_{tiller}$ increases, thus an increase in $CH_4$ emission (see Equation 16). Here it should be considered that the optimisation of plant related parameters depends on the plant species present in the wetland.





The posterior value for the porosity at the catotelm ($por_{cato}$) was observed considerably below the prior, indicating a more compact catotelm with less water (as we assume it is saturated). Change in water content will affect soil temperature slightly. This could have a dual effect for CH$_4$ such that it either increases the flux if the temperature increases in anaerobic condition, or decreases the flux due to compact peat. As described in Section 3.2.1, the $por_{acro}$ remained unchanged; hence no changes in acrotelm porosity occurred. The positive kurtosis observed in the PDF of this parameter indicates a well constrained single solution, and the negative skewness indicates a more probabilistic region below the posterior estimate.

The posterior value for $\lambda_{root}$ is estimated to be much smaller than the prior (more than 1/3 of $1\sigma$ of the prior estimate). This small posterior value for $\lambda_{root}$ indicates a low decay length of root biomass in the soil, means more of the decomposition and CH$_4$ production occurs in the acrotelm, and less in the catotelm. The emission of CH$_4$ produced mainly by peat decomposition in the acrotelm would be facilitated by a low posterior value for $\lambda_{root}$, with around 60% in the first layer of acrotelm followed by 22% and 8% in the second and third layers of acrotelm.

### 4.3 Posterior correlation estimates

This study conducted a detailed analysis of posterior parameter-parameter correlations, parameter-process correlation, and process-process correlation. The detailed discussion is given below.

#### 4.3.1 Posterior parameter-parameter correlations

Figure 8 provides an overview of the posterior parameter characteristics. The following discussion distinguishes between strong ($> 0.5$) and weak ($< 0.2$) parameter correlations. A strong negative correlation is observed between $Rmoist$ and $Rmoist_{anaerobic}$. The posterior estimate of the parameter values also shows an opposite tendency in these parameters (Table 6), indicating reduced CH$_4$ production in acrotelm and increased CH$_4$ production in catotelm. The parameter $CH_4/CO_2$ is negatively correlated to the parameters $Rmoist_{anaerobic}$, and to $\lambda_{root}$. This indicates a reduction in CH$_4$ fraction produced by decomposition in deep soil. Increase in tiller weight would add more organic carbon to the soil, which will result in a more compact peat accumulation in the bottom layers of soil with less porosity. This might be the reason for the negative correlation between $w_{tiller}$ and $por_{cato}$. $por_{acro}$ showed a very high positive correlation to $f_{air}$, which can simply be explained as more porous soil allows for more air in the soil. $w_{tiller}$ showed a strong positive correlation with $\phi_{tiller}$ and $r_{tiller}$ indicating an overall positive correlation among the parameters related to the plant-mediated transport. All the other parameters showed rather weak positive or negative correlations.

#### 4.3.2 Posterior parameter-process correlation

As described in Section 2.2, the total CH$_4$ flux simulated by LPJ-GUESS is calculated by summing up the component fluxes from diffusion, ebullition and plant-mediated transport. The following discusses (based on the Figure 7 ) the interactions between the optimised process parameters and the component fluxes.





**Moisture response, $Rmoist$ and $Rmoist_{an}$**

If larger the value of $Rmoist$, it would likely results in a faster soil carbon turnover time, which makes more carbon available

for $CH_4$ production, hence a slight increase in emission. The weak positive correlation and regression slope of $Rmoist$ with all the flux components could be due to this enhanced turnover time. $Rmoist_{anaerobic}$ had a positive effect on diffusion and a negative effect on plant-mediated transport. The positive effect is because of the same reason of enhanced carbon turnover time, but the negative effect is due to the low plant root abundance in saturated catotelm. Increase in $Rmoist_{anaerobic}$ contributed very less to the ebullition. This is most likely because of the negligible contribution of ebullition to the overall flux, having zero

contribution for most of the time.

**Methane/carbon dioxide ratio, $CH_4/CO_2$**

The very high positive correlation and regression slope of the $CH_4/CO_2$ parameter with diffusion and plant-mediated transport (which are the two diffusive pathways) indicates that a large part of the total emission of $CH_4$ are through these pathways. Higher the value of $CH_4/CO_2$, the more carbon is channelled into the $CH_4$ pool. Especially the plants with a larger tiller

radius after the optimisation are able to transport more $CH_4$ when there is more dissolved $CH_4$ available in the soil. The increase in ebullition is marginally less (smaller circle) than the other fluxes, most likely because ebullition is limited by the availability of gaseous $CH_4$, produced when the solubility reaches the maximum. However, the dissolved $CH_4$ is first emitted via diffusive fluxes; hence, there is very little $CH_4$ left in the gaseous phase for ebullition.

**Oxidation fraction, $f_{oxid}$**

The fraction of available oxygen utilized for $CH_4$ oxidation is determined by the parameter $f_{oxid}$. It showed a negative correlation to diffusion and ebullition and a slight positive correlation to plant-mediated transport. A decrease in diffusion and ebullition can be explained by a greater fraction of available oxygen used for $CH_4$ oxidation leading to less $CH_4$ emitted via ebullition. Significant decrease occurs in diffusion since the diffusive flux cannot circumvent the top layer, into which oxygen diffuses. Direct explanation of the increase in plant-mediated transport is hard due to the complex process formulation in the

model, but, it should be noted that the aerenchymas could transport a part of the oxygen deep down to the soil layers where it plays less of a role in oxidation, but contributes more to the total gas pressure, which can escalate the passive plant mediated transport to the atmosphere.

**Transport, $\phi_{tiller}$, $r_{tiller}$ and $w_{tiller}$**

As mentioned before the parameters $\phi_{tiller}$, $r_{tiller}$ and $w_{tiller}$ are positively correlated to each others. They are also positively

correlated (with a positive slope) to plant-mediated transport. These parameters could have two effects on the emissions: Having aerenchyma cells with more porous space, radius and biomass, on the one hand, enhances the $CH_4$ transport to the atmosphere, but on the other hand, through the same spacious aerenchyma cells, it is also possible for plants to transport more $O_2$ to the





soil. This enhanced O$_2$ transport to the soil could be the reason for the slight reduction in diffusion and ebullition observed in the cases of $\phi_{tiller}$ and $w_{tiller}$.

**615**  **Fraction of air and porosity, $f_{air}$, $por_{acro}$ and $por_{cato}$**

The slightly positive posterior value $f_{air}$ increased all the flux components with a comparatively larger effect on diffusion. As stated above, the diffusivity of CH$_4$ in air is four orders of magnitude larger than that in water, indicating fast and easy transport of CH$_4$ to the atmosphere. The slight increase in diffusion could be the direct influence of $f_{air}$ as this parameter is the main controlling parameter of this component flux in the model.

**620**  Similar to $f_{air}$, the $por_{acro}$ also had a posterior value that increased the fluxes from all components but with a rather low correlation. The reason for this is, as explained above, that a larger parameter value means a higher amount of air in the soil and hence more ebullition. In contrast, the $por_{cato}$ slightly reduced the ebullition. This could be because more water can potentially occupy the pores of permanently saturated catotelm which will indirectly affect ebullition through phase change and by affecting on soil temperature.

**625**  **Decay length, $\lambda_{root}$**

$\lambda_{root}$ played a key role in this optimisation. Figure 7 showed that $\lambda_{root}$ has a strong negative regression slope to diffusion and a weak positive regression slope to plant-mediated transport. The value got reduced considerably (higher than the value reported in Wania et al. (2010) and in Susiluoto et al. (2018)) after the optimisation and resulting in a much shallower soil profile for most of the root decay. As most of the peat decomposition activities are assumed to happen in acrotelm the reduction in **630** the magnitude of $\lambda_{root}$ facilitated diffusion, especially as it is the largest component. On the other hand, the plant-mediated transport got reduced due to the reduction in the root depth controlling parameter $\lambda_{root}$.

### 4.3.3 Posterior flux components

In Figure 12, the time series of process components are shown for the posterior mean estimate. In general, the optimisation of the model parameters leads to around 50% decrease in the production of CH$_4$ compared to the prior, with a considerable **635** reduction in plant-mediated and ebullition components, leaving diffusion as the dominant component. Diffusion reduced by around 30% and the plant-mediated transport reduces around 86 %. The low contribution of plant transport is mainly due to the low value of the root depth controlling parameter $\lambda_{root}$, which got reduced from 25.17 to a value of 10.58. This lower proportion of the plant-mediated transport is however surprising for a fen wetland site like Siikaneva with the greater aerenchymous leaf area throughout the growing season. The result is contradictory to the results obtained from optimising the model sqHIMMELI **640** (Susiluoto et al., 2018), in which the largest fraction of CH$_4$ is contributed by the plant-mediated transport. However, from the field experiments conducted at Siikaneva to estimate the plant-mediated transport, Korrensalo et al. (2022) has observed a smaller proportion of the ecosystem scale CH$_4$ efflux attributable to plant CH$_4$ transport in the Siikaneva fen site, which is well in agreement with the result we observed.



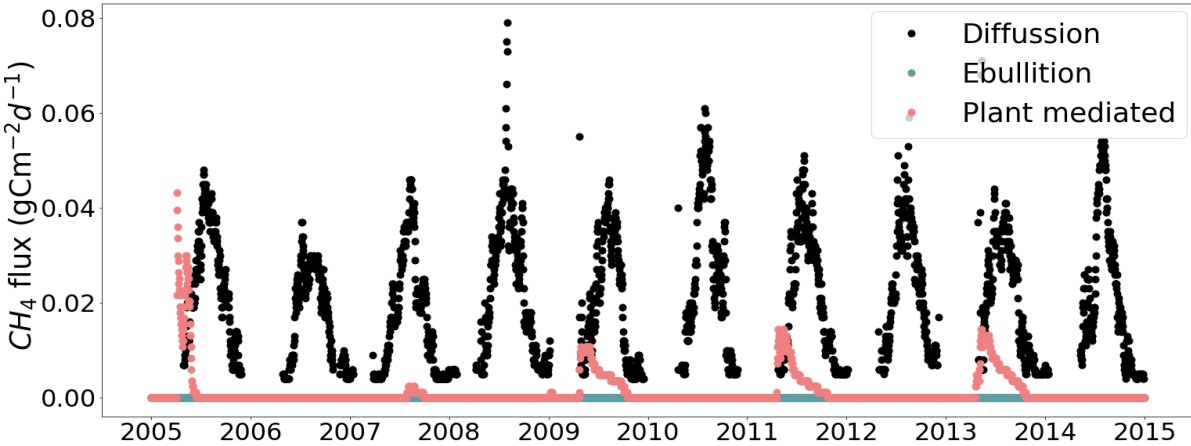

**Figure 12.** Time series for diffusion, ebullition, and plant transport using parameter values from the posterior mean estimate. The figure illustrates the changes in ebullition and plant-mediated transport following the optimization. A few outliers above 0.08 $gCm^{-2}$ on the vertical axis have been removed from the figure for the better visualisation.

The largest reduction, however, was for ebullition by around 92%. This result is not surprising since Wania et al. (2010),
who provide the basic foundation of the $CH_4$ model in LPJ-GUESS, also reported almost virtually no ebullition to the surface at several sites. Figure 12 shows that during the years 2008, 2010 and 2012, there is no ebullition estimated. Here, it should also be considered that the representation of ebullition in LPJ-GUESS is somewhat simplified as it is represented by a curve fitting equation for calculating the solubility, and the ideal gas law is applied to convert the volume of $CH_4$ per volume of water into the corresponding number of moles. Due to this lack of detail and its fast timescale occurrence (mostly depends on the
physical parameters such as temperature and pressure) and with no relevant parameters in the control vector, the optimisation could not alter the ebullition component directly. But on the other hand, the ebullition is indirectly controlled by parameters related to $CH_4$ production and transport when there is high saturated $CH_4$ available in the soil water, and thus the optimisation can change the ebullition component indirectly. The overall total of the observed $CH_4$ flux from Siikaneva during the period of 2005 to 2014 was 56.0 $gCm^{-2}$, and the prior estimate of the model was 98.5 $gCm^{-2}$ (Table 7). After the optimisation,
with the posterior mean estimate of parameter values, the model estimated flux of 53.5 $gCm^{-2}$ with an estimated posterior uncertainty of $\mp$ 4.82. This shows a reduced model-data error after optimisation with a difference of only 2.5 $gCm^{-2}$.

### 4.3.4 Posterior process-process correlation

After the optimisation, the air fraction in the peat got increased, which is likely the cause of the enhanced diffusion. Diffusion is estimated in the model based on the soil porosity and water, temperate and air fractions in the soil. Correlating the diffusion
to the ebullition showed a negative result, i. e. illustrating the dominance of diffusion over ebullition under more air in peat (see Figure 7b). A larger air fraction in the soil can also lead to an increase in plant-mediated emissions as the passive diffusion of air through the plant tissues depends on the amount of air in the soil/peat water (see Section 2.3.3). This can be seen in Figure 7b





**Table 7.** Total emission from flux components estimated from MAP posterior mean and prior parameter values for the optimisation time period. The unit is in $gCm^{-2}$.

| Component | MAP | Posterior mean | Prior | Observation |
|---|---|---|---|---|
| Diffusion | 49.5 | 49.6 | 70.7 | |
| Ebullition | 0.15 | 0.28 | 4.1 | |
| Plant-mediated | 3.5 | 3.7 | 23.6 | |
| Total | 53 | 53.5 | 98.5 | 56.0 |

as a comparatively high correlation between diffusion and plant-mediated transport. The increased tiller radius $r_{tiller}$ in plants increases the $A_{tiller}$ value (Equation 16), and hence also favours faster diffusion through the aerenchyma cells. Ebullition is

positively correlated to plant-mediated transport, indicating the occurrence of both these components when there is a high concentration of $CH_4$ in the soil. This occurs when the water table is located close to the surface and when there are more graminoids. An increase in plant-mediated transport of gases to the soil increases the net pressure imposed by the gases in soil/peat water, which likely leads to increased ebullition.

### 4.4 Model error and fit to the observation

The annual mean errors for the prior parameter values, MAP, and posterior mean values are shown in Figure 10 as one std. Except for ebullition, all the prior process components exhibited larger variances of the annual errors compared to the posterior estimates. The plant-mediated transport is the component with the largest error in the prior estimate. The posterior error estimates for this component showed nearly equal values with a slightly higher value for the posterior mean estimate. A similar pattern can also be seen for diffusion. In contrast to this, the MAP error estimate for ebullition showed a higher value compared

to the posterior mean error but interestingly also to the prior. The posterior mean error estimate for ebullition showed the lowest value.

The annual flux components mentioned above are illustrated in Figure 9a. It is clear from this figure that the prior process components had large inter-annual variance, especially for the first three years and last year. Considerable reduction in variance is observed for both the MAP and posterior mean estimates. The reduction of the variance observed in posterior estimates is

not proportional to the prior, but still, the posterior estimates showed comparatively high variance in the first and last years. In Figure 9b (as described in Section 3.2.4) the posterior mean estimate shows a comparatively high variance (w.r.t the MAP estimate) of the annual errors with a negative bias throughout the time period. In contrast to this, the MAP estimate showed a positive bias throughout the time period. Compared to the posterior mean estimate, the MAP estimate has considerably larger parameter values for the $\phi_{tiller}$ and $r_{tiller}$ which could possibly be interpreted as slightly more $CH_4$ emission through

the increased tillers of plants, hence the reason for the positive bias of the MAP estimate. Figure 10 also indicates a high percentage of annual plant-mediated emissions for the MAP estimate. The negative bias of the posterior mean estimate could





be due to the additional wintertime emission from the real-world wetlands, which is not captured in the model. In the model, the emissions start around early summer, once the soil is not frozen anymore. In addition, the large daily variability in the observations of the summertime fluxes is also not represented in the model. Overall the posterior estimates of the annual

fluxes are in good agreement with the observations leading to a small model-data mismatch for both MAP and posterior mean estimates.

### 4.5   Model inputs and uncertainty

After the optimisation, the model result showed a small underestimation of the cumulative emission between 2005-2014 around 2.5 $gCm^{-2}$. One reason for this mismatch could be the daily variations in the input climate data. The model is unable to repre-

sent peak emissions caused by the micro-environmental changes. As mentioned in Wania et al. (2010), the flux components are complex processes that depend on changes in many environmental factors. For instance, ebullition (one of the more complex $CH_4$ emissions processes in LPJ-GUESS as shown in previous sections) depends on volumetric content of wind and various gases. and hydrostatic and atmospheric pressure, but the model is not using them as forcing. Ebullition is also affected by the concentration of $CH_4$ and the density of nucleation sites, which are difficult to represent in the model. Apart from these

potential bias contributors, the CENTURY soil scheme and soil temperature and hydrology calculations used in the model also contributes to the uncertainty in the model predictions. Given these caveats, the small negative biases obtained for the posterior mean estimates when compared with the observation (see Figure 9 b) are reasonable when considering the quality and uncertainty of the input data used (see Section 2.1).

### 4.6   Optimised simulation from LPJ-GUESS

A detailed time series distribution of prior and posterior model simulations plotted against the observation is shown in Figure 11. The posterior model predictions were adjusted by the optimisation to capture the observation with considerable adjustment to the summer peaks. For example, the large peaks in the modelled emissions in 2005 and 2006, which largely contributed to the prior cost function, disappeared in the posterior emissions. In the following years, 2007 and 2008, the prior model simulations underestimated the observation, which also got corrected in the posterior. Also, the posterior emissions largely

capture the comparatively high peaks in the observations for the years 2010 and 2012, though the model still underestimated the observation. In 2013, the observations were high and the optimisation failed to capture this peak; rather, it tried to compensate for the underestimation by releasing a sudden high spike at the end of the summer that year. In winter months, the model simulated zero fluxes (as discussed before), whereas, the observations showed a small emission (around 8.3 % of the assimilated total), often with some small spikes possibly from the ebullition. This inability of the model to capture the wintertime emission

has contributed to the posterior model uncertainty and model data misfit.

As discussed in Section 4.3.4, the contribution of ebullition to the posterior estimate is comparatively negligible. Compared to the posterior, there were many emissions spikes observed in the prior estimate, especially during the beginning and the end of the summer months. Apart from these spikes the prior $CH_4$ estimates during the summer were a bit low in most of the years. The posterior estimate has considerably reduced these high spikes and adjusted the summer peaks to match the observation





better. On the other hand, while compromising with the summer peaks in the observation, the optimized parameter often failed to capture the abrupt high fluxes in the daily observation and simulated them at slightly wrong times. The spike shown at the end of 2013 is an example of such a mis-timing. This is likely to be caused by errors in the meteorological input data and missing wind and pressure.

It can be seen from the Figure 11 that the majority of the observations lie within the 95 % confidence interval of the posterior
estimate. Often the observation uncertainty overlaps the confidence interval except for the summer peak times of 2010, 2012 and 2013, in which the observation showed strong peaks compared to the average values. The few outliers in the observations are not captured by the model; these could likely be measurement artefacts and/or due to environmental forcing not considered here, again such as wind speed or air pressure.

## 5 Conclusions

This study marks the initial effort to optimize the model process parameters controlling the simulation of wetland $CH_4$ fluxes within the LPJ-GUESS model using the Rao-Blackwellised adaptive MCMC technique based on Bayesian statistics. The assimilation framework has been shown to be able to retrieve correct parameter values by performing a set of twin experiments. Furthermore, we used eddy-covariance flux measurement data from a boreal wetland to calibrate the LPJ-GUESS model parameters for a site-specific simulation. The results demonstrated that the fit to the observation of the $CH_4$ simulation of a
complex terrestrial DGVM like LPJ-GUESS can be systematically enhanced with a Bayesian parameter calibration. The results also showed that the modelled processes and the estimated parameters were well constrained by the observations leading to a substantial reduction in the posterior uncertainty of the simulated $CH_4$ emissions. The results of the re-sampling experiment indicated that there were no redundant processes in the model description, as shown by the parameter and process correlations.

The robustness of the assimilation framework developed in this study calls for further application of the framework using
observations from multiple sites in a simultaneous assimilation. Further validation of the framework's performance is neces- sary to confirm its applicability to other sites with diverse plant functional types and climatic conditions. The relatively strong roughness in the shape of the cost function observed in this study is expected to be reduced in a multi-site assimilation experi- ment, as has been observed by Kuppel et al. (2012), which would allow the retrieval of the global minimum of the cost function more easily. These further applications are beyond the scope of this paper and will be investigated in future studies.

*Code and data availability.* The AMCMC code and data used for this article is available at https://zenodo.org/record/7657117#.Y_NwPOxBy3I. The LPJ-GUESS model can be obtained here: https://web.nateko.lu.se/lpj-guess/. Currently, the model code is not completely accessible to the public. However, it has been provided to the editor and shared anonymously with the reviewers.

*Author contributions.* JTK designed the G-RB AM assimilation framework with the help from MS and JL. PM developed the $CH_4$ model in LPJ-GUESS based on the work of (Wania et al., 2010). JR, MS, PM and MR provided knowledge and advice about peatland $CH_4$ processes



and model parameters. MS and JL provided the supervision and technical advice with programming the algorithm and analyzing results. JTK prepared the manuscript with contributions from all co-authors.

*Competing interests.* The authors have the following competing interests: At least one of the (co-)authors is a member of the editorial board of Geoscientific Model Development.

*Acknowledgements.* We thank Jing Tang, Guillaume Monteil, Adrian Gustafso, Stefan Olin, and Johan Nord for the discussion of the com-
putational setup and the manuscript. The Institute for Atmospheric and Earth System Research (SMEAR-INAR) at the University of Helsinki is acknowledged for the CH$_4$ flux data collected at Siikaneva. We would also like to acknowledge the Finnish Meteorological Institute (FMI), Finland for open climate data availability.

This research has been supported by the Strategic Research Area: Biodiversity and Ecosystem services in a Changing Climate (BECC), Lund University,and is a contribution to the Strategic Research Area: ModElling the Regional and Global Earth system
(MERGE). BECC and MERGE are funded by the Swedish government.





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
