# Peer review of "Optimising CH4 simulations from the LPJ-GUESS model v4.1 using an adaptive MCMC algorithm"

_Geoscientific Model Development, 2022_

## Author Response (AR1)

Dear Julia,

I wish to express my gratitude for your assistance in navigating the publication process of our paper, 'Optimizing CH4 Simulations from the LPJ-GUESS Model v4.1 Using an Adaptive MCMC Algorithm'. Your efforts in identifying suitable reviewers for our paper are greatly appreciated.

We are appreciate to have received constructive feedback from both reviewers. We believe that their careful assessments and insightful comments have significantly contributed to the enhancement of our manuscript's quality.

I'm pleased to inform you that we have now addressed the feedback from both reviewers and have submitted our responses (author's comments). In line with the reviewers' recommendations, we have made significant revisions and restructured the text, ensuring that all relevant details remain in the paper. Furthermore, we have included a detailed description of the technical aspects of the G-RB Algorithm in the paper and have added additional details and plots to both the paper and the appendix to address some of the comments.

Once again, I would like to express our gratitude to you and the dedicated reviewers for the time and effort in helping to improve our manuscript.

Best regards,
Jalisha

*Note: The copy of both AC replays are attached below*

**Anonymous Referee #1**

*We are very thankful to the Referee for the constructive comments. In the following, we have addressed the Referee's comments. The manuscript has also been revised accordingly.*

*Note: The reviewer comments are referred to in black font type throughout the texts, and the authors' responses are referred to as blue font.*

This is an interesting study on the optimization of the methane emission parameterization in LPJ-GUESS using flux measurements. The Bayesian optimization method that is used provides a more objective basis to derive unknown parameters in the model than the initial approach taken by Wania et al when designing the parameterization. The authors rightly regard the currently reported step as an initial effort towards this goal, as measurements from only one site are used. However, it serves the purpose of documenting a promising method and provides a reference for future work. A few issues remain to be resolved, as explained below.

1. The text is quite long :a stronger focus on the most important findings would help.

We have the same comment also from referee #2. Considering this, along with the editor's suggestion to keep the relevant details somewhere in the paper, we have made the following changes to the paper.

- Only the main model processor is now included in the paper, and we have moved a significant portion of the model description to the appendix. We have retained the model description because we believe it is important when discussing the results.
- We have rewritten the discussion in accordance with the comments of the referee #2, resulting in a significant reduction in text length. The discussions on parameter correlations, process parameter correlations, and process-process correlations have been combined into a single short section with more clarity.
- At the request of the referee #2 we have moved the technical details of the MCMC framework from the appendix to the main paper.

2. The discussion of parameter correlations is sometimes difficult to follow, and needs further clarification as explained below.

As mentioned above, the discussion of parameter-parameter and parameter-process components has now been combined, reduced, and simplified.

The optimized model shows significant non-random misfits to the data, in particular regarding interannual variability, in a way that cannot easily be explained by data uncertainty. Some further discussion or analysis is needed for the reader to understand what it means. Is the state vector incomplete?. 6. Is there any correlation between the misfit and driving environmental variables (temperature, water availability, vegetation productivity, …). Currently this issue is kind of glossed over in a way that calls the sophisticated new mathematical machinery that is presented into question.

Many thanks for pointing these things out. We have observed that in the real data experiment, for some years (2010, 2012, and 2013), there are systematic

underestimations of emissions. Since in the twin experiment we do not such systematic errors we have added a new figure in the main paper, Figure 3, showing the time series estimates of the twin experiments in Scenario 1. The simulations, using four sets of posterior parameter values obtained from the twin experiments, are plotted against the twin observations used. Our assumption for these systematic errors in the real data experiment is that these are likely caused by a model structural error, i.e. we think there are inherent structural issues in LPJ-GUESS. While the methane module within LPJ-GUESS is relatively comprehensive when compared to many other similar models, the model's process description and parameterisation remain incomplete. For instance, in the real world, wind plays a crucial role in methane emissions and its atmospheric concentration. However, in LPJ-GUESS, wind speed is set to zero for modelling convenience, which presents a significant limitation. Similarly, the lack of representation of atmospheric pressure and simplified implementation of the ebullition component is also limitations.

Furthermore, the state vector is indeed incomplete. LPJ-GUESS is a complex model encompassing numerous ecosystem processes. These processes are interconnected, and there exists a large set of parameters directly or indirectly linked to methane fluxes. There are two primary reasons for selecting a limited number of parameters within the methane model of the system: Firstly, our aim was to assess the feasibility of the framework we developed around LPJ-GUESS. Consequently, we opted to avoid the complications arising from an excessive number of dimensions and parameter correlations. Representing the indirect parameters can be intricate, as they may depend on other fluxes or model components. For instance, LPJ-GUESS's soil module is intricately connected to the Century model, featuring ten soil compartments. Introducing a parameter related to soil temperature or water table into the framework would necessitate accounting for the intricacies of the Century model. And secondly, this would require the inclusion of additional observations, such as soil temperature profiles or water table profiles otherwise the optimisation problem would be heavily underdetermined. This would significantly increase the complexity of the problem, exceeding the scope of this paper.

Considering the suggestion to examine the correlation between the misfit and driving environmental variables (Air temperature, Precipitation, and Short Wave Radiation), we have now examined these correlations and added the plots and results in the appendix (see S3 of the appendix), as well as to the discussion in Section 4.4 (Model Inputs and Uncertainty) of the main paper. We have also examined the observed and simulated soil temperature and water table depth. See S4 of the appendix and section 4.5 (Optimised simulation from LPJ-GUESS). Despite explaining the misfit with the observed correlations, it might be necessary to conduct future studies to include soil temperature, NEE, Water Table Depth, and other factors. Again, incorporating additional observations and parameters would exceed the scope of this paper.

3. Section 2.2. provides 'brief descriptions' of the main processes in the model of a bit more than 8 pages. I understand that there are many components and that this is the consequence of describing them all briefly.

Yes, the methane model description in LPJ-GUESS is relatively extensive. We found it challenging to discuss the parameters and their correlation without the description of the underlying model equations and processes. However, as mentioned above, we have now moved a significant portion of the model description to the appendix.

As I understand it, there is a detailed description of the current model setup in Gustafson (2022), but details are also in McGuire (2012). It raises questions about where the reader should turn to for a full description of a particular component of the model and if the information about the model that is provided here is only repeating that information in a summarized form or also includes actual modifications. It may be better to summarize the main processes with references to papers that describe the details for each of them and only show detailed equations later to explain the parameters that have been selected for optimization. Right now, something in between a full and summarized model description is provided, from which it is difficult to infer if this model version is the same or not than what was used before, and where one should turn to for details about any specific component of it.

It is true that the detailed description can be found in Gustafson (2022), and a few details in McGuire (2012). The description in McGuire (2012) cannot be considered a reference for the model processes, as it is incomplete and differs in the representation of the processes. The actual model description can be found in Wania (2009a, 2009b, and 2010), originally written for LPJ-WHYME (This has been mentioned in the paper). This code was later adapted and modified for LPJ-GUESS with some adjustments and additional details (see the Section 2.2 of the paper). This is one of the reasons for including the detailed model description in this manuscript. Also, when we initially submitted the manuscript to GMD, Gustafson (2022) had not yet been published, and there was no citable reference for the actual methane description in LPJ-GUESS. Additionally, the LPJ-GUESS model code is now publicly available, including the description of the methane module, which was not the case earlier.

As mentioned above, we have moved a significant portion of the model description to the appendix. While we appreciate the suggestion to summarise the main processes and provide detailed comments, we found it very challenging to include only a portion of the processes, as we believe it might confuse readers and make it difficult to follow the discussion. Since the model code is now publicly available and will continue to be updated with new releases, we recommend that interested parties refer to the latest versions of the model.

4. Section 4 presents the parameters that are selected for optimization. For the parameters from Wania et al (2010) it is clear why they were selected, but not for the seven other parameters. They were selected for their high SI value, but how was that determined? How many parameters were tested?

We tested 16 parameters directly associated with the methane module of the model (based on expert opinion) and ultimately incorporated 11 of them into our study due to their high Sensitivity Index (SI). We acknowledge the mistake of not including the eliminated parameters in the paper. Figure 2 in the paper has now been modified to include the less sensitive parameters. A brief description of these eliminated parameters can be found in the paper, Section 2.3, as follows:

"We considered five of the seven parameters Wania et al (2010) tested in their sensitivity analysis (two parameters related to the root exudate decomposition are not used in LPJ-GUESS) together with eleven other parameters used in LPJ-GUESS. Based on their high SI values we chose eleven of them for the optimisation (Figure:02, Table:01). Among the five eliminated parameters ag_frac is the fraction of ANPP (Annual Net Primary Production) used to calculate number of tillers, DT_min is the minimum temperature (degree C) for heterotrophic decomposition, por_org is the porosity of organic material,

C_con is the carbon content of biomass and U_10 is the possible constant value of wind speed at 10m height."

In my understanding, the CH4 flux variability at Siikaneva is largely controlled by temperature, water table, and vegetation productivity. It is not so clear to me that the chosen parameters reflect the main uncertainties in each of these.

We agree that the variability in CH4 flux at Siikaneva is primarily influenced by temperature, water table depth, and vegetation productivity. As mentioned above to limit the number of parameters and keep the problem's dimension smaller (to test the usability of the framework, which we think is the primary objective of the paper), we selected parameters directly connected to the CH4 module of the model. The parameters directly linked to temperature and water table depth are not fully represented by our chosen parameters. On the other hand, since some Plant Functional Types (PFTs) are exclusively added to the wetland CH4 model, we considered many parameters related to vegetation productivity, such as w_tiller, r_tiller, phi_tiller, and lambda_root. Therefore, it's important to acknowledge that while our parameter selection reflects key elements influencing CH4 flux, further investigation with more parameters and fluxes may be necessary to comprehensively capture the main sources of uncertainty.

Even though the temperature and humidity response of the emission may arise from underlying model parameters, it would nevertheless by useful to know the sensitivity of the model to those drivers and if the optimized model behaves different than the prior. Quantifying those sensitivities would also facilitate the comparison with other studies and models.

Considering the suggestion, to examine the sensitivity of the LPJ-GUESS to the input variables before and after the optimisation, we have now conducted a sensitivity analysis. Sensitivity is estimated by increasing the air temperature by one degree and short wave radiation and precipitation by 10% of the original site values. The estimates of sensitivity consistently decreased after optimisation. However, the change was relatively small, with a reduction of less than 10% for all three variables. The details are given in the S5 of appendix with a mention in the paper (Section: 4.4) as below:

"The results of the sensitivity study indicate that both the prior and posterior model estimates are significantly sensitive to the input variables. However, the posterior model estimate exhibited a reduction in sensitivity especially for short wave radiation and precipitation (see Appendix S5 )."

5. In section 2.5 the parameter optimization framework is explained. From the description I conclude that it is iterative only on the prolongation of the Markov chain, but not in the starting point of the optimization process. For the Twin experiment different starting points were used, but not for the real data experiment. It raises the question if the convergence is similar using synthetic and real data. Some further discussion is needed about this.

We appreciate your valuable comment, and it highlights an important aspect of our study. The twin experiment was designed to assess the applicability of the G-RB AM framework within LPJ-GUESS. To test the convergence capability of the framework, we initiated each twin experiment with different starting points for the parameters. Once we confirmed that the framework could converge consistently regardless of the initial parameter values, we proceeded with a single set of real data experiments. Given the valid question

regarding starting from different points in the real data experiment, we revisited some of our preliminary analyses that were not included in this study. This allowed us to verify that the parameters indeed exhibited convergence behaviour towards the posterior parameter values obtained in our final analysis (though we observed somme issues of equifinality).

For the resampling experiment it is unclear how the random selection was done. Is it drawn from the full posterior error covariance matrix? In this case, the correlations that are examined are not sensitivities of one parameter to another, but error correlations, so the sensitivity of the error in one parameter to the error in another.
The discussion that follows in the results section is very difficult to follow without knowing exactly how to interpret parameter to parameter correlations. If the aim is to determine the sensitivity of a flux component to a certain parameter, then why not perturb only that parameter relative to the optimal solution?

1000 sets of parameters are randomly selected from the posterior distribution (the last 25% of samples from the MCMC chain). Interpreting the MCMC from a Bayesian perspective, samples from the last part of the chain correspond to samples from the posterior parameter distribution, and will thus be a representative draw of "reasonable" parameters given the observations. Strong correlation in the posterior should be interpreted as combinations of parameters that produce similar outputs, and thus represent combinations that are not well constrained/identified by the available observations rather than as parameter errors. These parameters are then used to run the model and simulate the total CH4 flux and its components. The outputs from all 1000 simulations are then analysed to understand the correlations among processes and the influence of each parameter on the flux components and total emissions. In response to the feedback regarding the difficulty in understanding the posterior correlation matrix, we have revised the text to make it more straightforward.

**SPECIFIC COMMENTS**

Line 84: 'The air temperature …' Correlation of data on what timescale? For rainfall the relevant timescale may be a bit longer, for which the correlation could be higher.

Ten years of meteorological data were collected from both Siikaneva and Hyytiala. In the data collected from Siikaneva, out of the expected 4017 daily data points, 793 data points were missing for Temperature, 1326 were missing for Precipitation, and 19 were missing for shortwave radiation, see Figure. 1.

[Figure]

(Figure : 1)

Equation 10: '..CH4/CO2..' I thought the model kept track of oxygen in the soil. Then why is CH4/CO2 treated as a constant in the model? Line 214: the same question as the previous for f_oxid.

The model keeps track of oxygen in the soil. In wetlands, the oxygen in the soil is determined by the fraction of air in the soil, and the degree of anoxia is defined as 1 - (Fair + fair). Were, Fair is the fraction of air in the soil layers and fair is the fraction of air in peat (Wania et al., 2009a) (see Sections 2.21 (CH4 production) and S1 in Supplement for details). The fraction of plant roots in the soil influences the transport of oxygen into and out of the soil. So, the molar ratio of CH4 to CO2 production is weighted by these two factors to estimate CH4 production. Depending on the oxygen availability, a part of this CH4 will get oxidised, denoted as $f_{oxid}$. The CH4 to CO2 ratio is an adjustable parameter in the model due to the wide variation in the molar ratio of CH4 to C02.

Line 326: Why is the cost function called a loss function? (here for the first time but also in other places)

Technically both the cost function and loss function are correct terms. Both terms are used to describe a mathematical function that quantifies how well a model's predictions match the observation. In our manuscript, we initially used both terms interchangeably. However, we have since updated the terminology to use 'cost function' exclusively.

Line 353: Why are the prior values set to Ztrue if the performance of the twin experiment is determined by the ability of the MCMC to recover Ztrue?

The cost function is a combination of model fit and deviation from the prior. For the twin experiment we have taken the prior-mean  equal to the true parameters to ensure that the true minimum of the combined cost function coincides with the minimum of only the model fit part. If the prior-mean does not equal the true parameters it is conceivable that the combined minimum could deviate from the minimum for the model fit part. For a real world application the prior implies a weighting between prior knowledge and guesses about the parameters and information obtained from the current observations. Here, the difference in prior variance and observational variance as well as the amount of data will determine how that weighting is done. The prior sensitivity and selection of suitable prior means and variances is an interesting question, but outside the scope of this study.

Line 370 – 375: Which criteria are used to classify as 'large' or 'low' std?

If the standard deviation of the distribution is greater than 20% of its total range (ie, total range of the parameters), it is considered as a large standard deviation. This 20% threshold is based on expert opinion and is discussed in detail in Section 2.7 of the paper (Parameter value estimation ) as:

"To be more precise with the estimation, for the posterior parameter distributions appeared multi-model if the std of the distribution is greater than 20% of its total range, we classified them as poorly constrained. The edge-hitting parameters are the ones that cluster near one of the edges of their prior range (Braswell et al., 2005). "

Figure 3: which of the optimization chains is chosen here? The same question for for S2:2.

It is of the experiment one of the scenario1. We have mentioned it in the Section 3.1 (Twin experiment using G-RB AM) as:

"The resulting PDFs of the experiment 1 after the 'burn-in' are represented in Figure 4."

And,
"Posterior parameter correlations of the experiment 1 shown in Figure S2:1 are given in S2:2. "

Figure S2:1 The choice of initial points seem biased low for most parameters. Is this by accident? Could the biased convergence of lambda_root and CH4/CO2 be explained by this?

This happened partially by accident. However, In principle, a good initial guess can significantly speed up the convergence of the MCMC algorithm. Therefore, we were randomly searching for better starting points independently for each parameter. This situation arose when we encountered large differences between the proposed and previous cost values, making it difficult to calculate the logarithm of the difference. To address this issue, we decided to implement tempering. We then sought a few sets of parameters with a moderately good cost values to begin with.

[Figure]

(Figure: 2)

We examined our other experiments that we have not used for this study to see if it is possible to explain the biased convergence of lambda_root and CH4/CO2 by the initial points chosen. Figure 2 shows experiments with different starting points than the one we used for this study. The figure indicates a trade off between CH4/CO2 and lambda_root, where when one converges, the other one is staying away from the true value. It should be noted that when lambda_root increases, the total emissions tend to strongly decrease, and when CH4/CO2 increases, the total emissions tend to strongly increase, see Figure : 2 above.

Line 410: It is not clear from the setup if the flux measurements are perturbed with random noise or not. It seems not to be the case as the expectation is that the solution converges to Ztrue (rather than to Ztrue within the posterior uncertainty margin). It might explain why chi2 is systematically below 1. This should be explained clearer, as well as the implication for the representativeness of the Twin Experiment for the performance of the optimization using real data.

Many thanks for pointing this out. No, we have not perturbed the flux measurement with any random noise. The expectation is that the parameters converge to the Ztrue values, without any model or observational error, and that is definitely a reason why the $\chi^2$ is systematically below 1. A reduced $\chi^2$ value systematically below 1 for different twin experiments might suggest an issue with the representation of uncertainty or errors in both the model and observation. As the twin experiments here are run under the assumption of an 'idealised model'—meaning the model perfectly reproduces the observations without any errors or uncertainty—and 'error-free data,' where the data perfectly represents the environmental conditions without any systematic or measurement errors, it's expected to have $\chi^2$ values systematically below 1. In this case, we expect the model to fit the observations without considering any uncertainty. Also, the $\chi^2$ value is highly sensitive to the number of observations and parameters. With 3650 observations in Scenario1 and 41610 observations in Scenario 2, and only 11 parameters, this can lead to low $\chi^2$ values. So, considering these facts, we believe that having low $\chi^2$ values for the twin experiment doesn't necessarily affect the framework's ability to be set up for the real data experiment. We have now mentioned this in Section 4.1.

Table 6: Why has the cost function value not been translated into its corresponding chi2 value, which would provide more insight in the optimization process (e.g. compared with that of the twin experiment)

Thank you for bringing this to our attention. In response, we have made the following change to the Section 3.2.3 to address this concern:

 "After the optimisation, the cost function value was reduced to 2959.8 (reduced χ2=3.82) with the MAP estimate of parameters and to 3002.6 (reduced χ2=3.88) with the posterior mean estimate of parameters."

Line 534: '… which is around half the prior model estimate' What do you mean here?

We are sorry for the confusion. The total magnitude of the observation is around half of the prior estimate. The details are in Table 6, i.e. the magnitude of total observation is 56, whereas the total prior is 98.5. For the clarity we now have changed the text to:

"Here, it should be considered that, in this study, assimilation aims to reduce the magnitude of the prior CH4 simulation to minimise its misfit with the observed data, which is nearly half of the prior model estimate (see Table 6)."

Figure 12: '.. The figure illustrates the change in … optimization' But the results shown are the posterior contributions to the flux, not the difference with the prior, right? In that case it is not the 'change' that is shown.

Yes. That is correct. Many thanks for noticing it. It is not the changes, rather the component decomposition of the total flux. We have corrected it now in the paper.

Figure 12: What explains that plant mediated transport is only important early in the season? You would expect the plant-mediated transport to become more important as the growing season progresses.

This is true. But what the posterior resulted is that the plant mediated emission is observed only in some years. And in the years it is present, it seems more pronounced early in the summer of these years. We assume this could be due to the reduced value of the root depth-controlling parameter lambda_root, which decreased from 25.17 to 10.58 after the optimisation.

Line 635: What signal in the measured fluxes becomes easier to fit by the model if the ratio between diffusion and plant mediated transport changes? Can we understand why this happens?

Predicting this is challenging. In LPJ-GUESS, processes are represented in a complex manner, connecting environmental, vegetation, and climatic variables. In cases where the prior model overestimates real fluxes (as we had in this study), the assimilation attempts to reduce the modelled flux by decreasing emissions from the components. If we could utilise complete state vectors and additional related fluxes, the posterior should derive the most plausible proportions of each component. However, if the state vector and process description are incomplete, the optimisation attempts to find the most viable solution by adjusting emissions from different components. One way to achieve the accurate proportion of fluxes from the optimisation could be by adding observed data from all the flux components, however, this is not practical due to the lack of observational data for the flux components.

Table 7: Is the reduction of the posterior flux compared with the prior only driven by the observations 2005 and 2006? In the other years it is not clear that the flux should go down.

We apologise for the lack of clarity. It is the total reduction obtained by summing all the ten years. We now have changed the caption of the figure to make it clear (Please note that the table number is changed from 7 to 6 now).

Figure 11: Is there a reason why the posterior model has difficulty reproducing the observed inter-annual variability? It does a good job on the multi annual mean flux. That it cannot reproduce short-term variability is understandable and probably not so critical. However, it would be useful to understand the drivers of inter-annual variability and why the model has difficulty capturing them.

Many thanks for the comment. Yes, there are some systematic misfits observed in the posterior estimate. The wetland module is connected to several biogeochemical cycles and environmental variables. Hundreds of parameters contribute to it. As we have mentioned before, there could be many reasons for this, such as an incomplete state vector, incomplete process representation, variability in the model inputs, etc. We have mentioned this in the context of discussing the correlation between observed model inputs and posterior CH4 residuals as an answer to the comment 2 above. We intend to extend future studies to include soil temperature, NEE, WTD, and other factors.

**TECHNICAL CORRECTION**

Line 7: delete 'are'

Line 76: '… measurements'

Line 85: '… therefor …'

Line 102: '.. are are …'

Line 106: in the current formulation the 10m water depth seems to refer to the snow layers

Line 129: '.. ((wtd)) ..' Somehow wtd always appears in parentheses.

Line 276: '… divdied …'

Line 525: '… resulted three …'

584: 'A larger value ... would ... result'

Many thanks for pointing out the technical errors they are truly valuable. We have now corrected them in the revised paper.

**Anonymous Referee #2**

*We are very thankful to the Referee for the constructive comments. In the following, we have addressed the Referee's comments. The manuscript has also been revised accordingly.*

*Note: The reviewer comments are referred to in black font type throughout the texts, and the authors' responses are referred to as blue font.*

The study "Optimising CH4 simulations from the LPJ-GUESS model v4.1 using an adaptive MCMC algorithm" presents a parameter calibration methodology and example case targeting methane emissions simulated by the LPJ-GUESS model. The goal of this study, parameter optimization for simulations of complex processes is important and timely in the field of ecosystem modeling.

However, the manuscript is very long, with detailed descriptions of model structure and individual parameter responses to optimization. The study would be greatly improved by clearer and more high-level descriptions of the key points, a more explicit evaluation of the parameter optimization methodology, and fewer less-relevant details.

For me, this would include:

1.  Less description of the LPJ-GUESS model since Wania et al 2010 includes much of the same information. It is mostly important to highlight key differences between the version of the model used here and previously published model descriptions.

We have the same comment also from the referee #1. And the comment is addressed as the response to the first comment of referee #1. The key differences between the methane modules of the LPJWHyMe and LPJ-GUESS can be found in Section 2.2 in the main paper and S1 of the appendix.

2. A better description of the G-RB AM framework, since this is the primary subject of the study. Some of the information in the supplementary could be moved to the main text, and the algorithms described in Andrieu and Thoms 2008 could be presented/ summarized here.

The primary reason we opted not to include the technical details in the main paper is due to our concern that it might lead to further increase in its length. Also, the algorithms we used, a mixture of the Rao-Black-wellised Adaptive Metropolis (AM) algorithm and the Global Adaptive scaling AM algorithm are comprehensively described in Andrieu and Tomas (2008). However it is true that the primary objective of the study is the development and testing of our framework. Taking into careful consideration your valuable feedback, we have relocated the description of G-RB AM framework to the main paper.

3. Twin Experiment: this is a very useful test of the parameter optimization methodology, but the interpretation of the results of this experiment could use some improvement. To my eye, the methodology did not do a great job at recovering the parameters in an application that is much much easier than one involving real field data. For example, the CH4/CO2 is a very key parameter in the LPJ-GUESS methane model, since it directly tunes methane production. Why was there such difficulty recovering this parameter?

You are right that in principle the twin experiment should be able to recover the parameters completely. To verify the robustness of twin experiment, now we have added a new plot to the paper (Figure: 3, explained in Section 3.1 and Section 4.1) that displays all the posterior time series from our four twin experiments in Scenario 1 in comparison to the twin observation we used. The figure shows how effectively the optimised parameters, following the twin experiment, capture the structure of the twin observation and the majority of its spikes. The broader parametrisation challenges we observed in the real data experiment, that most likely are related to issues with the parametrisation, are not observed in the twin experiment. For example, it can be noted from this figure that none of these experiments display the systematic underestimation of the flux observed in the real data experiment.

In general, we agree that the twin experiment did not completely recover the true values of these parameters. We mentioned the parameters CH4/CO2 and lambda_root particularly in the paper because these were the two parameters that were not recovered to their true values at least once in the posterior distribution. The LPJ-GUESS model is very complex and extremely non-linear in its behaviour. When we mentioned the good convergence of all parameters except for CH4/CO2 and lambda_root, we were considering this fact about the model, hence the possibility of different parameter combinations yielding similar results for the same problem (equifinality).

And, in addition to the parameters mentioned in L386, I see poor convergence for Rmoist, fair, porcato, and maybe foxid too in Figure S1. Why is the optimization framework encountering these difficulties, and what would be needed to improve it?

This could be attributed to the following problems :
1. The high dimensionality of the problem and limited variability in the data for its resolution. For instance, regarding the parameter CH4/CO2, one reason for the poor recovery could be that we are only optimising the CH4 component, and the proportion of CO2 is significantly off. If we incorporate both CH4 and CO2 fluxes to address both sides of the ratio, it will lead to a more accurate convergence.
2. The high degree of non-linearity and complexity in the model, which results in equifinality.
3. Limited dimensions in the data. Achieving complete convergence may require different types of observations with climatic and geographical variations. Here the model is not fully constrained by the limited observations.

A possible solution we are considering is that a single type of flux from a single site might not be sufficient for identifying these parameters due to the high degree of non-linearity in the model. Therefore, additional sites with spanning a large climatic variability and/or different species of fluxes (for example, in addition to assimilating CH4 observations also assimilating CO2 observations) from the same site may be necessary. However, a challenge in this approach would involve formulating cost functions for different observations in a manner that properly weights them to equally represent the information

without diminishing its significance, and these considerations are beyond the scope of this paper.

Also, what are the 4 experiments in Figure S1?

We have considered two scenarios for the twin experiment to test the identifiability of the parameters under different conditions. Scenario 1 with a shorter temporal scale from 2005 to 2014 (10 years), and scenario 2 with a longer temporal scale from 1901 to 2015 (115 years). Four sets of chains for both scenarios with a chain length of 100,000 iterations were analysed. In each set of the scenarios, the optimisation started from a different initial point in parameter space randomly selected from their prescribed ranges. This has been described in Section 2.6, and Section 3.1 in the paper.

4. Real Data experiment: This is the central result of the study, so I think that some summary statistics (RMSE, R2) of model performance before and after parameter optimization should be included in the abstract.

This is a very valid comment we somehow neglected. We now have added the line "The experiment using real observations from Siikaneva resulted in a reduction of RMSE from 0.044 gC/m2/d1 to 0.023 gC/m2/d1 along with a 93.89% reduction in the cost function value" to the abstract.

Further, I think the discussion of this experiment should focus more on the final fit of model to data, and less on the responses of individual parameters to optimization.

We have now merged most of the discussion about the parameter-parameter correlations and process parameter correlations into one section (Section 4.2.1). We have also considerably revised the Sections 4.1 and 4.4.

Discussion of the various causes of model/data mismatch (which is quite large in 2010, 2012, and 2013) should be strengthened, and should include some discussion of incomplete or incorrect process representation (e.g. methane production (eq. 10) is fairly simplistic). Some comparison of measured vs modeled soil temperature, VWC etc. would also be quite helpful, and may help to explain some of the model/data mismatch.

Thank you, this is an important comment about the misfit we have received from both the referees. Carefully considering this, we have now revised Section 4.4 (Model Inputs and Uncertainty) to address the misfits observed in the above-mentioned years. We have also discussed the possible issues with the process representation and parameterisation.

None of the parameters we optimised in this study are directly related to the model's soil temperature or the water table depth. Therefore, comparing observed versus modelled soil temperature or water table depth may not provide substantial insights in this context. However, in response to the comment suggesting the usefulness of such comparisons for explaining certain model-data discrepancies, we now examined the observed and modelled soil temperature (at 5 m depth) and water table depth to see if there is any relation. We did not observe any major differences between the prior and posterior of these components. However, a significant mismatch with the observations is observed. This mismatch is likely the reason for the zero simulated emission estimate during the winter, and it could partially explain some of the observed misfits. Still, it may not be able to explain the systematic misfit observed. The result can be seen in the Appendix S4 and

is discussed in Section 4.5 (Optimised simulation from LPJ-GUESS). The observed data used are mentioned at the end of the Section 2.1 (Siikaneva wetland and measurements).

According to the suggestion of referee #1 we have also examined the model's input data (Solar Radiation, Temperature, and Precipitation) in relation to the simulated flux residuals (S3 in the Supplement) and have discussed this in Section 4.4.

And, why not directly compare simulated vegetation mediated fluxes to those measured by Korrensalo et al (2022)?

Comparing the individual components of flux simulated by LPJ-GUESS with actual observations can be a valuable approach for refining the optimization process and gaining insights into simulation variabilities. However, the data provided by Korrensalo et al. (2022) is not technically usable in this context for the following reasons:

1. We wanted our approach to keep more general and transferable to other site. It could be hard to find observation of different component fluxes from other sites to assimilate.
2. They provided daily data for plant-mediated CH4 transport from the Siikaneva fen for a limited time period, specifically May to October (a duration of 5 months) in the year 2014. To achieve meaningful comparisons and interpretations, a more extensive dataset covering inter annual variability is required.
3. The measurement of plant CH4 transport rate, expressed as mg CH4 per gram of plant dry mass within the chamber per day (mg CH4 g−1 day−1), was conducted using custom-made cylinder-shaped chambers and then scaled up to the ecosystem level. This upscaling process introduces certain uncertainties that need to be addressed.
4. We assimilated the entire CH4 flux into the model and examined its individual components to gain a better understanding of the variabilities in the simulation. While it is possible to assimilate all the components of the fluxes individually to provide stronger model constraints, this would require additional data for all the flux components, which falls slightly outside the scope of this study.

5. One of the main difficulties of parameter optimization is that it is computationally intensive, and the optimization algorithm used in this study is designed to reduce the number of chains, and therefore the computing time, needed for convergence. For others interested in applying a similar approach in different contexts, information about computing time would be very useful - i.e. approx. how much computing time is needed for one model run and how much for the entire optimization procedure. Estimated improvements on running time vs an unoptimized MCMC approach would also be of interest.

It is indeed true that parameter optimization methods in general, and especially the MCMC approach, are computationally intensive and time-consuming. In this specific case, each LPJ-GUESS model simulation took approximately 9 seconds to finish. Consequently, for the 100,000 iterations, it consumed nearly 250 computational hours. However, it's important to take into account that in this study, the model has been running for a single site, and the computational speed is highly dependent on the performance of the processors being used.

We have now added these details in Section 2.6 (Experiment design) as "We have designed a set of twin experiments and a real data experiment. For both the twin and real

data experiments, we generated MCMC chains with a length of 100,000 samples. MCMC approaches are computationally intensive and time-consuming. In this study, each model simulation took approximately 9 seconds to complete (using an AMD Ryzen Threadripper processor). As a result, for the 100,000 iterations, it consumed nearly 250 computational hours. However, it should be noted that this study involves the model running for a single site, and the computational speed is highly dependent on the performance of the processors being used"

5. The paper would benefit from a general discussion of the merits and shortcomings of this method of parameter optimization supported by the experiments performed in this study.

Many thanks, we agree with this opinion. To address this, we have now added a subsection in the discussion, please see Section 4.6.

**Minor comments**
1.  L42: repetition: "to model their
2. Table 3: Description of poracro and porcato
3. Supplementary: tabel -> table
4. Figure 11: why isn't the 95% confidence interval of the simulations zero in winter, since the model can't simulate methane fluxes then?

Thanks for pointing out the technical error, they are truly valuable. We have now addressed them in the revised manuscript. The confidence interval we used here provides a range of values within which we believe the true population parameter is likely to fall based on our sample data and a specified level of confidence. We used this to understand the uncertainty associated with our estimates.

---

## Author Response (AR2)

**Reply to the editor:**

*I remain concerned that the identical twin tests do not recover the parameters. I was disappointed that you did not resolve this in your revision. If identical twin tests fail, it is not usually appropriate to proceed with using real data. Instead the experimental design should be adjusted until the test succeeds. In such a situation I would probably initially try reducing the number of parameters being estimated in order to make the problem more tractable. Did you try this? The reviewer does still think that the methodology and results presented are useful to the community, but there needs to be much more discussion of the limitations. I hope you can see their second review on the manuscript overview, and will be able to provide a revision in due course.*

We would like to extend our sincere gratitude for your assistance in guiding us through the publication process of our paper 'Optimising CH4 Simulations from the LPJ- GUESS Model v4.1 Using an Adaptive MCMC Algorithm' in GMD. We appreciate your response and the decision to call for a second revision, which we believe would be instrumental in refining the manuscript.

With regard to the issue you raised in your last letter, we agree that, in principle, the twin experiment should be able to recover all the parameters completely. We acknowledge (now also in the manuscript, section 4.6) that the twin experiment in this study did not fully recover the true values of certain parameters, especially the parameter CH4/CO2. As mentioned in the manuscript and in responses to the reviewers, LPJ-GUESS is a highly complex and highly non-linear model. There is a significant likelihood of different parameter combinations yielding similar results for the same problem (known as equifinality). Based on our understanding, we assume that the equifinality problem might be inherent to the parametrisation in LPJ-GUESS rather than the optimisation algorithm. We provided some indications of this in our response to Review 1, demonstrating the offsetting behaviour of lambda_root and CH4/CO2. In two prior twin experiments, we observed that when one of these parameters converged, the other one consistently exhibited offsetting behaviour and did not converge.

The high dimensionality of the problem and limited variability in the data for its resolution could be another reason for the poor convergence of some parameters. Considering the high degree of non-linearity in the model, we believe that relying on a single type of flux from a single site for the twin experiment might not be sufficient for identifying all the parameters. Therefore, additional sites spanning a wide climatic variability and/or different species of fluxes from the same site may be necessary. For instance, in the case of the parameter CH4/CO2, one reason for the poor recovery could be that the study only constrain the CH4 component, while the part of CO2 is significantly off. Incorporating both CH4 and CO2 fluxes to address both sides of the ratio can lead to a more accurate convergence. These considerations, however, are beyond the scope of this paper, and further studies are currently in progress in these aspects. Another reason for the incomplete convergence could be that the model is not fully constrained by the limited dimensions in the parameter space. Achieving complete convergence may require incorporating additional parameters from different modules of the model that represent various processes.

In response to your suggestion to redo the twin experiment with a reduced number of parameters, technically, it is possible to decrease the number of parameters and rerun the experiment. However, this approach introduces some challenges. A reduced number of parameters would lead to the derivation of an entirely different subsystem of the model, which may not necessarily represent the complete system examined in this study. In the current setup, the parameters co-vary simultaneously after each accepted iteration, resulting in a set of parameters that undergo offsetting and adjustments throughout the exploration process. If we were to reduce the number of parameters, the system would perform the same operations in a more simplistic/low-dimensional manner and potentially recover the original parameters. The problem here is that it won't have much scientific relevance if we use a subsystem or even multiple subsystems separately to have all the parameters included in twin experiments that recover the true values, but then use the full system with all the parameters for the real data optimisation. Alternatively, if we choose a new subsystem for the twin and real data experiment, this would necessitate redoing the entire experiment and rewriting the entire paper with a different set of parameters. We are concerned that this is not a minor revision, but rather turns into substantial revisions that would ultimately change the entire paper. Another significant issue here is the large computational time involved.

Instead of redoing the twin experiment now, we would like to explore the possibility of addressing the reviewer's comment and explicitly tackling the issue of the non-converging behaviour of some parameters, especially of $CH_4/CO_2$. This is also along the lines of the reviewer's request. In response, we have now added a detailed description of this issue in Section 4.6. We hope this will sufficiently address your concerns and the reviewer's suggestions.

Apart from this, we kindly request your attention to a minor change we made to the paper at this stage. We have altered the name of the MCMC algorithm we developed from G-RB AM to GRaB-AM for the ease of use and readability.

Once again, we would like to express our gratitude to you and the dedicated reviewers for your and their time and effort in revising our manuscript. We eagerly await your positive decision and the publication of our work.

**Reply to the referee #2**

*I still have some concerns and confusion about the CH4/CO2 parameter. Your sensitivity analysis shows that CH4 flux is most sensitive to this parameter, yet the twin experiment fails to recover it (or even really explore the full space of this parameter). When real observations are used the CH4/CO2 parameter is estimated to be considerably lower than the prior, but this parallels the incorrect underestimation of this parameter by the twin experiment, eroding trust that the optimisation framework is improving simulated fluxes for the right reasons. Given that many readers may have similar concerns and confusion, I would suggest incorporating some of the discussion in your review response (i.e. the impact of model equifinality, future work on simultaneous optimisation of CH4 and CO2) into the paper. Perhaps this could fit in an expanded Section 4.6 that includes discussion of future work.*

Many thanks, your observation is true. Though the parameter CH4/CO2 in the experiment shown in the paper has been observed to tend to approach the true value, it has failed to converge to the true value. On the other hand, the parameter has shown convergence to the true value in our other experiments, as shown and described in Revision 1 (Section: Specific Comment). We assume this parameter might exhibit an offsetting behaviour to the parameter lambda_root.

Considering your valuable suggestion and the possible confusion readers might have, we have now extended Section 4.6 with a more detailed explanation of the nonlinearity of the model, the possibility of equifinality in this context, and the need for more species of flux integration in the system. On the other hand, we believe that redoing a twin experiment with different starting points, longer chain lengths, or different sets of parameters would require a tremendous amount of work and would ultimately result in significant changes to the entire paper (please see the reply to the editor above).

---

## Author Response (AR3)

**Reply to the editor:**

many thanks again for taking your time to handle our manuscript. Frankly speaking, we are a bit puzzled by your persistent decision asking for major revisions and redoing the identical twin experiments for the following reasons and would request you to reconsider your decision.

First, the Referee #2 classified our manuscript in three out of the four categories as 'excellent' and in the fourth category as 'good'. Furthermore, she/he did not ask for any additional experiments but for 'incorporating some of the discussion in your review response' into the manuscript. We exactly did that in our last revision.

Second, we disagree that the identical twin experiment has not worked. As explained in our response, due to the high complexity and non-linearity of LPJ-GUESS, we expect equifinality issues, and that is exactly what we encountered with the identical twin experiments (as explained in the response as well as in the manuscript). We do see, however, when comparing the prior and posterior time series of the simulated CH4 emissions with the simulated observations that the posterior time series does capture the time series of the simulated observations much better than the time series using the prior parameter values (see Figure 1 here and also Table 4 in the manuscript listing the prior and posterior cost function values). Hence, we believe there is value in the identical twin experiments. This is also acknowledged by the referees.

[Figure]

Figure 1: Time series of simulated observations together with prior and posterior CH4 emission values from one of the twin experiment. The prior is shown in purple, posterior in blue and the twin observation in black

We also note that our study is not the only one encountering equifinality issues (or more generally problems) in retrieving the true parameter values from identical twin

experiments. Many similar studies working with high-dimensional Bayesian problems and non-linear models both in the Earth Sciences but also other fields have reported similar equifinality and parameter correlation issues, e.g. Hargreaves and Annan (2002), Apte et al. (2008), Annan and Hargreaves (2010), Long et al. ( 2010), Santaren et al. (2014), Lamminpää et al. (2019), just to name a few. Here, especially the study by Santaren et al. (2014) is comparable to ours as it also concerns model parameter optimisation against greenhouse gas flux observations, but it should be noted that we have achieved a much better parameter retrieval with the methodology employed in our study.

Considering your suggestion in your previous letter to investigate if the twin experiment can recover the true parameter values in a simplified, low-dimensional problem, we have conducted two sets of twin experiments with different sets of parameters. In the first set of parameters, we included the parameter CH4/CO2, and in the second set, we included the parameter lambda_root. These are the two parameters that are offsetting each other in the optimisation process. The results are given below in Figures 2 and 3. From these figures, it is clear that our algorithm is well capable of retrieving true parameters in a low-dimensional problem.

[Figure]

Figure 2: Four sets of twin experiments with six (incl. lambda_root and excl. CH4/CO2) out of the eleven parameters (optimised in the manuscript) started at different points in the parameter space. The straight black line in each panel indicates the true parameter values used to generate the identical twin observations.

[Figure]

Figure 3: Four sets of twin experiments with five (incl. CH4/CO2 and excl. lambda_root) out of eleven parameters (optimised in the manuscript) started at different points in the parameter space. The straight black line in each panel indicates the true parameter values used to generate the identical twin observations.

Finally, regarding your suggestion to include observational uncertainty in the identical twin experiments, we have already incorporated observational uncertainty to estimate the cost function.

Best regards
Jalisha

**References:**

1. Annan, J. D., and J. C. Hargreaves. "Efficient identification of ocean thermodynamics in a physical/biogeochemical ocean model with an iterative Importance Sampling method." Ocean Modelling 32.3-4 (2010): 205-215.
2. Apte, A., C. Jones, and A. M. Stuart. "A Bayesian approach to Lagrangian data assimilation." Tellus A 60.2 (2008): 336-347.
3. Hargreaves, J. C., and J. D. Annan. "Assimilation of paleo-data in a simple Earth system model." *Climate Dynamics* 19 (2002): 371-381.
4. Lamminpää, O., et al. "Accelerated MCMC for satellite-based measurements of atmospheric CO2." Remote Sensing 11.17 (2019): 2061.
5. Long, K. J., ,S. E. Haupt, and G. S. Young. "Assessing sensitivity of source term estimation." Atmospheric environment 44.12 (2010): 1558-1567

6.  Santaren, D. et al. "Ecosystem model optimization using in situ flux observations: benefit of Monte Carlo versus variational schemes and analyses of the year-to-year model performances." Biogeosciences 11.24 (2014): 7137-7158.

---

## Author Response (AR4)

Dear Julia,

Thank you very much for your prompt response. We sincerely appreciate your feedback on our twin experiments. In response to your comment, we have included additional content in the paper (Sections 4.1 and Section 4.6) to address and acknowledge the potential impacts of the lack of convergence on the subsequent real data experiment.

Furthermore, we are grateful for bringing to our attention the error in the discussion of the IDT observational uncertainty in the paper. The miswritten text was a result of collaborative input from various co-authors based on different experimental contexts. We have rectified this mistake in the revised version of the paper.

While you rightly suggested that fixing a couple of parameters might have rendered the experiment more manageable, it is important to note that, given the experimental nature aimed at testing the efficiency of the GRaB-AM algorithm, we opted to present the paper in its current form. We plan to explore different parameter combinations and further enhance the algorithm in our future studies.

Once again, thank you for your valuable insights and suggestions.

Best regards,
Jalisha